# Rapid evolution of *Klebsiella pneumoniae* biofilms in vitro delineates adaptive changes selected during infection

Greta Zaborskytė[1,3], Patrícia Coelho [1,2], Marie Wrande [1] & Linus Sandegren [1,2] ✉

Biofilm formation facilitates infection by the opportunistic pathogen *Klebsiella pneumoniae*, primarily via indwelling medical devices. Here, we explore the adaptive evolution of classical pathotype *K. pneumoniae* in surface-attached biofilms through experimental evolution that mimics catheter-associated infections. We observe rapid convergent evolution that alters or abolishes capsule production, modifies the fimbrial adhesin MrkD, or increases the production of fimbriae and cellulose via upregulated c-di-GMP-dependent pathways. We show that multiple aspects of biofilm formation, including early attachment, topology, surface preference, and extracellular matrix composition, are affected in a mutation-specific manner. However, evolutionary trajectories and resulting phenotypes show strain-specific differences, illustrating the importance of genetic background on biofilm adaptation. Additionally, changes in acute virulence are linked to the underlying genetic change rather than the overall biofilm capacity. Identified adaptive changes conferring hypermucoviscosity or affecting c-di-GMP-related regulatory pathways overlap extensively with those previously identified in clinical UTI and wound isolates, confirming biofilm as an important selective trait in vivo.

Biofilm growth is an inherent component of opportunistic infections[1]. Indwelling medical devices that support biofilm growth often facilitate the transition from colonization to infection[2,3]. *K. pneumoniae* of the classical pathotype (cKp) mainly causes hospital-acquired urinary tract infections (UTI) or ventilator-associated pneumonia via urinary catheters or endotracheal tubes[4–7]. The success of opportunistic cKp as a nosocomial pathogen is often studied from the perspective of its multidrug resistance[8], whereas its capacity to form biofilms that enable both patient colonization, infection, and survival in hospital settings remains understudied. Some genetic determinants, such as the *mrk* cluster, which encodes type 3 fimbriae in *K. pneumoniae*, have been characterized in detail with respect to molecular mechanisms and involvement in different types of biofilm[9–11]. However, the roles of gene networks in cKp biofilm dynamics are far from fully understood.

During both colonization and infection, bacterial populations undergo selection to adapt to various host niches[12]. Increased biofilm capacity is often found as a within-host adaptation in prolonged chronic infections, for example, by *Pseudomonas aeruginosa* and other bacteria in cystic fibrosis lungs[13,14]. Since biofilm communities embedded in an extracellular matrix are much more recalcitrant to antibiotic treatment and immune factors[1], they do not experience the same host-related bottlenecks as non-biofilm populations. Therefore, biofilms are often maintained for extended periods and can serve as reservoirs of bacteria that cause infections in other parts of the body. Furthermore, due to spatial constraints, biofilm growth forms distinct microenvironments that support further diversification between individual bacteria within the biofilm[15,16]. This is reflected in long-term within-host adaptations during biofilm-associated infections, where

[1]Department of Medical Biochemistry and Microbiology, Uppsala University, Uppsala, Sweden. [2]Uppsala Antibiotic Center, Uppsala University, Uppsala, Sweden. [3]Present address: Ineos Oxford Institute for Antimicrobial Research, Department of Chemistry, University of Oxford, Oxford, UK. ✉e-mail: linus.sandegren@imbim.uu.se

populations diversify in many aspects important for survival within the host, such as iron scavenging or antibiotic resistance[13,17,18]. Therefore, from an eco-evolutionary perspective, changes in biofilm capacity play a crucial role in shaping the overall pathoadaptive landscape at infection sites. To elucidate the molecular mechanisms behind such adaptations, experimental evolution in different biofilm model systems has proven a powerful tool[19,20]. Contrary to gene inactivation studies, experimental evolution avoids the bias of loss-of-function mutations, instead allowing selection of a broader spectrum of genetic variants under the given conditions[21].

Given that cKp frequently relies on surface interactions during hospital-acquired infections, studying the evolutionary changes that influence biofilm development can yield valuable insights into understanding its pathoadaptivity. While cKp can asymptomatically colonize the human gut for extended periods, the infections caused by this bacterium are rarely chronic. However, during an extensive clonal cKp hospital outbreak, we recently observed a repeated mutation-mediated switch to a chronic, rather than acutely virulent, state at infection sites that involved catheterization[22]. In line with this, we also identified increased biofilm formation on catheter-like surfaces as a repeatedly selected adaptive phenotype, primarily in patients with urinary tract infections[22]. Therefore, the question arises as to what extent such adaptations are selected due to the presence of an abiotic surface or are co-selected with other phenotypes that affect virulence.

Here, we explored the evolutionary trajectories and the underlying genetic networks that lead to increased biofilm formation in three clinical cKp strains, including the clone from the hospital outbreak, and compared them to the evolutionary changes selected within patients during the outbreak. Mutants with increased biofilm capacity were rapidly selected during short-term experimental evolution in an in vitro biofilm system mimicking catheter surfaces. Genotypically, there was extensive parallelism among the strains and independent lineages. Single nonsynonymous mutations were sufficient to drastically alter phenotypes in diverse, yet highly specific, mutation- and strain-dependent ways. We also demonstrate that biofilm-increasing mutations may lead to potential trade-offs in other features relevant during infection, such as sensitivity to innate immune factors. Notably, the in vitro selected changes strongly resemble those selected during within-host evolution, providing insights into why such changes can be advantageous during infection.

## Results

### Increased biofilm capacity is rapidly selected on catheter-like surfaces

To explore how classical *K. pneumoniae* evolves increased biofilm formation on abiotic surfaces, we performed experimental evolution using the FlexiPeg biofilm model system[23]. Multiple parallel bacterial populations were passaged on individual pegs over six 48 hour cycles, during which biofilms were formed, disrupted, and re-inoculated onto fresh pegs (Fig. 1a). To assess the influence of genetic background on evolutionary trajectories, we used three clinical cKp pathotype strains: the index isolate DA14734 (ST16, KL51, OL3γ) from a large clonal hospital outbreak in Uppsala, Sweden[22,24,25], the UTI isolate C3091[26] (ST14, KL16, OL2α.1), and the respiratory isolate IA565[27] (ST105, KL102, OL2α.2) that encodes an additional plasmid-borne type 3 fimbriae *mrk* cluster. Each strain was passaged in 32 independent lineages on three surface types: uncoated pegs, silicone-coated pegs, and pegs coated with silicone and fibrinogen from human plasma to resemble catheterization. Fibrinogen has been shown to accumulate on inserted catheters due to the local inflammatory response[28] and promote biofilm formation in several pathogens[29,30] including *K. pneumoniae*[6].

Initial biofilm-formation varied significantly by strain and surface type (p < 0.0001, two-way ANOVA), with C3091 consistently forming the largest attached populations (CFU/peg) across all surfaces (Fig. 1b). Consequently, while the endpoint population size increased

substantially with up to 4-log higher CFU/peg for the IA565 strain on silicone, the effects were more moderate for C3091, and 3/6 lineages of DA14734 went extinct on this surface (Fig. 1c and Supplementary Fig. 1). Biofilm formation was generally more robust on uncoated pegs, which have a rough surface[23], and in the presence of fibrinogen (Supplementary Fig. 1). Consistent with the role of extracellular matrix (ECM) as a main biofilm component[31], we observed a substantial increase in the accumulation of biomass on the pegs and aggregations in the wells over the course of cycling even when population sizes remained stable (Fig. 1d). Changes in colony morphologies on agar plates (morphotypes) have been frequently observed in other species during biofilm evolution[15,19,32]. Here, distinct morphotypes appeared in 40 of 96 lineages (Fig. 1e, f, and Supplementary Data 1) with a significant strain-dependent bias on uncoated pegs (Fig. 1f). Some lineages produced biomass extending >10 cm after lifting the peg lid (Fig. 1d and Supplementary Video 1), suggesting increased mucoviscosity. Such evolved IA565 colonies produced classical >5 mm "strings" in the hypermucoviscosity string test[33], whereas colonies of C3091 and DA14734 were instead tightly adhered to the agar (see videos 2-5 available on figshare [https://doi.org/10.6084/m9.figshare.29816147]). We also observed translucent colonies, indicative of capsule loss[34,35], and two lineages developed distinct wrinkly/rugose morphotypes (Fig. 1f), which are uncommon for *K. pneumoniae* but are associated with biofilm-enhancing mutations in other bacteria[15,19,32,36]. Together, these findings demonstrate that *K. pneumoniae* can rapidly diversify to enhance its biofilm-forming capacity when exposed to catheter-like surfaces, with outcomes strongly influenced by strain background and surface properties.

### Parallel genetic adaptations drive enhanced biofilm formation and mirror clinical selection

To identify genetic changes underlying increased biofilm formation, we performed whole-genome sequencing on selected clones and populations from lineages with increased biofilm capacity (total: 105 clones and 18 populations; see "Methods" and Supplementary Data 2). Mutations ranged from single nonsynonymous substitutions to combinations of up to seven mutations per clone (Fig. 2a). Multi-mutation genotypes were more common in the IA565 strain (p < 0.0005, Fisher's exact test), and in clones evolved on silicone or silicone with fibrinogen compared to uncoated pegs (Fig. 2b). Still, overall, biofilm capacity in multi-mutant clones was comparable to that of single mutants (p = 0.3819, two-way ANOVA; Fig. 2c). The overall number of unique mutated targets in clones originating from IA565 was almost twice (n = 36) that of C3091 (n = 18) or DA14734 (n = 16), but there was no significant bias across different surfaces (p = 0.3671, Fisher's exact test) and no consistent bias in specific gene or intergenic region (Supplementary Fig. 2). General point mutation rates (measured as rate of resistance to rifampicin) ranged between $3 \times 10^{-8}$ to $2 \times 10^{-9}$ per cell per generation for the three strains and were the same between planktonic growth and biofilm growth (Supplementary Fig. 3). This indicates that none of the strains have a mutator phenotype.

The capsule locus was the most frequently mutated target, with mutations found in 40 of 96 lineages. Notably, 33 of these affected *wzc*, a tyrosine autokinase essential for Wzy-dependent capsule synthesis and export in Gram-negative bacteria[37–39] (Fig. 3a and b). The second most common target was the type 3 fimbrial adhesin MrkD (20/96 lineages), followed by the MrkJ phosphodiesterase (12/96 lineages) involved in the turnover of the signaling molecule c-di-GMP[40]. In IA565, an uncharacterized gene within a putative type VI secretion system (T6SS) cluster (here named *spyo*) repeatedly acquired the same mutation, T361S (Fig. 3b), warranting further investigation in a separate study. Aside from a single mutation in a hypothetical plasmid gene in IA565, no plasmid-borne mutations were detected.

Since DA14734 is the index isolate from a large hospital outbreak, we compared the genetic changes selected during in vitro evolution

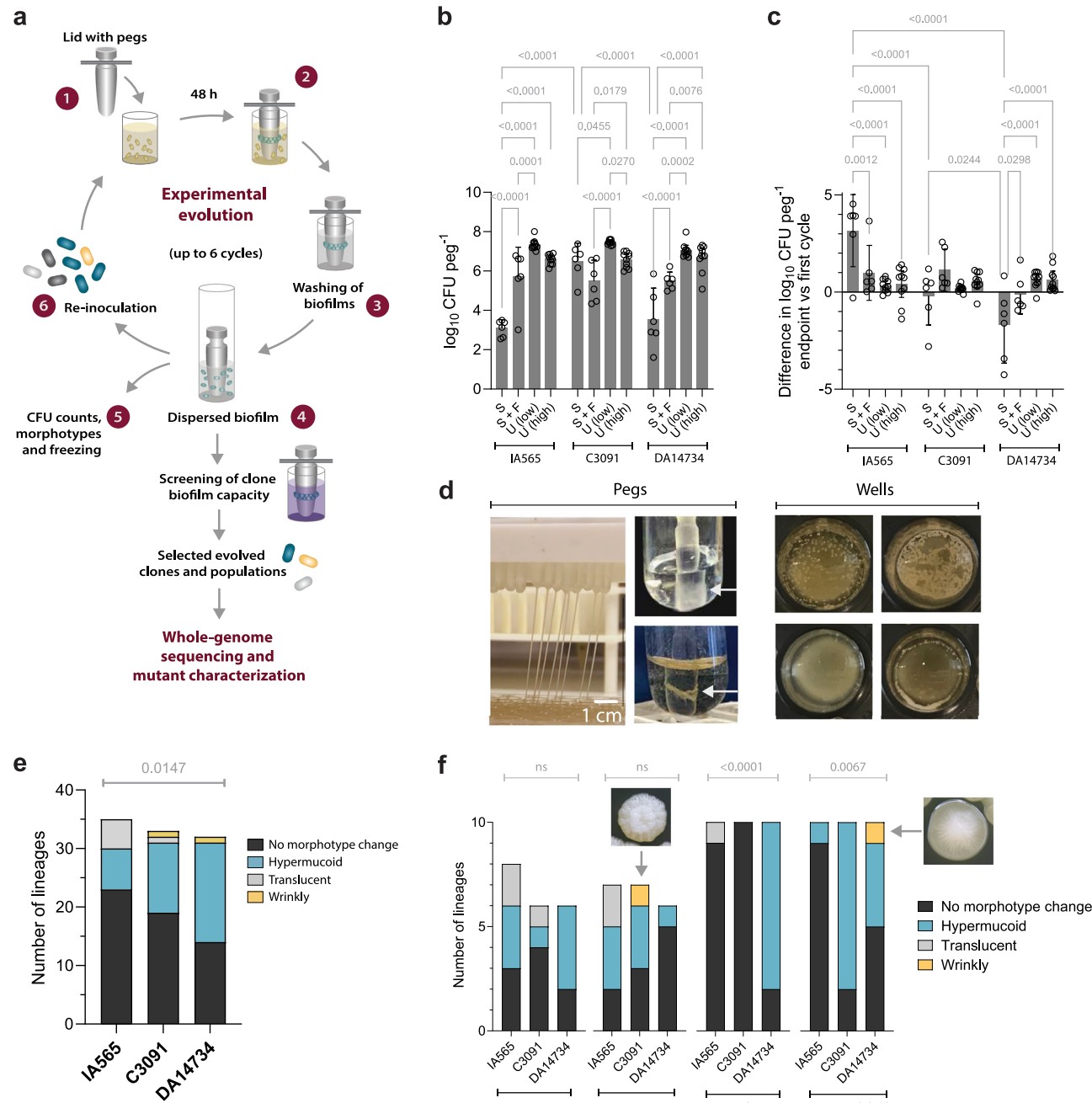

**Fig. 1 | Fast convergent evolution of increased biofilm formation in different *K. pneumoniae* strains. a** Scheme of experimental evolution of *K. pneumoniae* on the FlexiPeg device. (1) Diluted cultures were added to a 96-well plate and allowed to form biofilms on pegs for 48 h, (2) with a transfer to fresh medium after 24 h. (3) Pegs with attached biofilms were washed to remove planktonic and loosely bound bacteria, and (4) biofilms were dispersed by vortexing. (5) CFU/peg and morphotype frequencies in each lineage were assessed after each cycle. (6) Part of the harvested biofilm population was used to re-inoculate new biofilms. **b** Population sizes on pegs in the first cycle for all strains. Bars show means with 95% CI, and each datapoint represents an independent lineage, with *n* = 6 for each strain on silicone (S) and silicone with fibrinogen (S + F), and *n* = 10 for each strain on uncoated (U) pegs. Uncoated low and high refer to the size of the population transferred (0.25% and 25%, respectively) during cycling on pegs without silicone coating. Significance assessed using a two-way ANOVA followed by Tukey's multiple comparison test; two-sided *p* values exceeding significance (<0.05) are shown. **c** Change in CFU/peg

at the end of the experimental evolution. The CFU/peg of the first cycle was subtracted from the endpoint for each lineage. Bars show means with 95% CI, and each datapoint represents an independent lineage, with *n* = 6 for each strain on silicone (S) and silicone with fibrinogen (S + F), and *n* = 10 for each strain on uncoated (U) pegs. Uncoated low and high refer to the size of the population transferred (0.25% and 25%, respectively) during cycling on pegs without silicone coating. Significance assessed using a two-way ANOVA followed by Tukey's multiple comparison test; two-sided *p* values exceeding significance (<0.05) are shown. **d** Examples of biofilm phenotypes on pegs and in the wells during the cycling. **e, f** Quantification of evolving lineages with respect to morphotype (colony morphology) changes during cycling. Uncoated low and high refer to the size of the population transferred (0.25% and 25%, respectively) during cycling on pegs without silicone coating. Two wrinkly colony morphologies in (**f**) are shown as inserts next to their origin lineages (yellow). Significance assessed using Fisher's exact test; two-sided *p* values are shown. Source data are provided as a Source Data file.

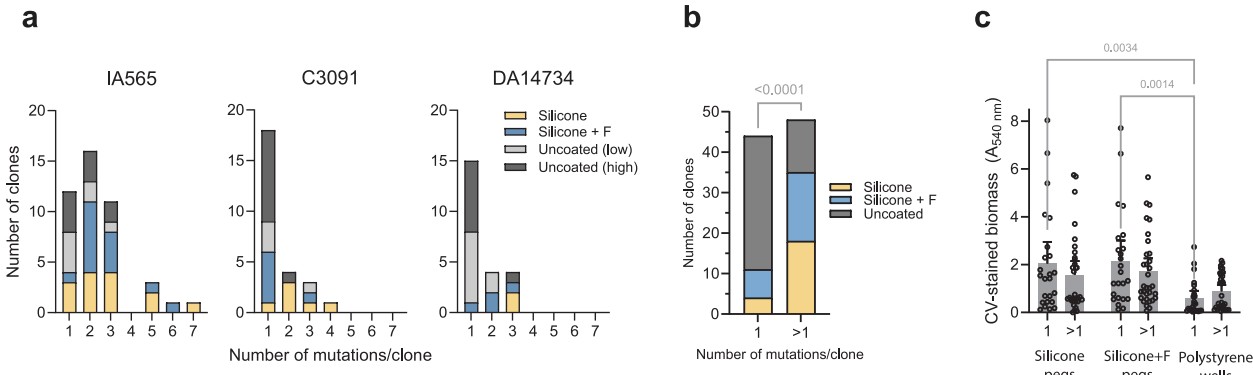

**Fig. 2 | The influence of surface and strain background on mutational characteristics. a** Histograms showing the number of mutations in individual evolved clones with increased biofilm formation. **b** Distribution of clones with single mutations or mutation combinations in lineages evolved on different surfaces. Statistical significance was assessed using Fisher's exact test; the two-sided $p$ value is shown. **c** Biofilm capacity of sequenced clones with unique single ("1", $n = 27$) or combined mutations (">1", $n = 32$) on silicone-coated pegs, silicone pegs with

additional fibrinogen coating, and in polystyrene microtiter plates. Biofilms were grown for 48 h and stained using crystal violet (CV). Bars show means with 95% CI, and each dot represents the mean of 4 biological replicates per clone. Comparisons were made using two-way ANOVA to determine the effects of the number of mutations and biofilm formation conditions; two-sided $p$ values exceeding significance are shown. Source data are provided as a Source Data file.

**Fig. 3 | Functional grouping and overlap of mutations repeatedly selected in vitro and in patients. a** Schematic illustration of the main overlapping mutational targets. Descriptions are given under the respective section for particular mutations. **b** Numbers in columns IA565, C3091, and DA14734 indicate how many independent lineages from all cycling rounds combined had clones with the respective mutation. The column "Outbreak" refers to the number of independent isolates that had the respective mutation during the outbreak at Uppsala University Hospital[22] Isolates from the outbreak are marked as colonizing (C) when isolated from fecal screening, and as urinary tract (U), blood (B), and wound (W) when isolated from infection sites.

with those found in isolates from patients[22]. Strikingly, there was a substantial overlap in functional targets (e.g., capsule, type 1 and 3 fimbriae, cellulose), specific genes, and even identical amino acid substitutions between biofilm-enhancing mutations selected in vitro and those found in outbreak isolates (Fig. 3b). The overlapping targets and mutations were predominantly present in isolates from infection sites ($n = 32$), especially UTIs (n = 27), and significantly less common in colonizing isolates ($n = 7$) (two-tailed binomial test, $p < 0.0001$). Importantly, *wzc* missense mutations, which we previously identified as adaptive in the host, were exclusively found in isolates from infections (UTIs and wounds), suggesting niche-specific selection pressures. Collectively, these findings demonstrate that *K. pneumoniae* evolves biofilm-enhancing genotypes through highly convergent pathways across strains and surfaces, and that these adaptations closely resemble those selected during human infection.

## Morphotype selection dynamics and biofilm capacity are surface- and context-dependent

Whole-genome sequencing of selected clones revealed the genetic basis for the different colony morphotypes that emerged during biofilm evolution (Fig. 4a). Hypermucoid lineages producing long threads of biomass from the pegs in liquid, including "string-test-positive" IA565 colonies and "stuck-on-agar" colonies from C3091 and DA14734, were consistently associated with single missense mutations in the *wzc* gene. Translucent morphotypes resulted from inactivation of the initial capsule glycosyltransferase WbaP or the phosphatase Wzb, or were derived from the hypermucoid morphotype by additional frameshifts in *wzc*. While *wzc* and capsule-loss mutations could co-occur with other mutations, such clones still maintained their hypermucoid and non-mucoid morphotypes, respectively. The wrinkly morphotype in C3091 was linked to a missense mutation (A203V) in DL426_RS09305, an uncharacterized GGDEF domain protein likely involved in c-di-GMP signaling. Notably, we found the same mutation and wrinkly morphotype in a UTI isolate from the hospital outbreak, in the homologous gene I6N99_05140[22] (Fig. 3b). The second in vitro-evolved wrinkly morphotype, found in DA14734, was temperature-dependent (wrinkly at 37°C, parental-like at 30°C), and mediated by an S325T mutation in the global transcription terminator Rho. This mutation is located near one of the positions (324, in the R-loop) identified as crucial for transcriptional termination[41].

Biofilm capacity was dependent on the particular mutation and surface for all morphotypes (p < 0.0001, two-way ANOVA) (Fig. 4b). The hypermucoid *wzc* mutants formed robust biofilms on silicone with fibrinogen but showed almost total lack of attachment to polystyrene (Fig. 4a, b). However, they formed abundant "fluffy" non-attached biomass when grown in microtiter plates, potentially explaining why a previous study reported Wzc mutants as non-biofilm forming[35]. In contrast, mutants affecting fimbrial adhesins - with or without loss of capsule - formed biofilm on polystyrene, suggesting broader surface compatibility. The wrinkly Rho$^{S325T}$ mutant formed extremely robust biofilms, especially on silicone with fibrinogen at 30°C, where its colony morphotype reverted to parental-like. Likewise, the wrinkly DL426_RS09305$^{A203V}$ mutant was a particularly good biofilm former on fibrinogen-coated surfaces, the condition under which it was selected. Among mutants with parental-like morphotype, RcsD$^{G217R}$ in IA565 formed one of the most robust biofilms on silicone-coated pegs both with and without fibrinogen. RcsD is a histidine phosphotransferase in the Rcs envelope stress response system, which is involved in capsule synthesis, biofilm formation, and host infection[42,43]. The G217R mutation, located in the periplasmic region of RcsD, likely results in loss-of-function, as a constructed deletion of *rcsD* also led to increased biofilm formation. Overall, mutants exhibited greater cross-correlation and total biomass production on silicone and silicone with fibrinogen compared to polystyrene (Fig. 4c).

To track the evolutionary trajectories during selection, we quantified the different morphologies in the harvested biofilm populations

at each cycle and observed how different morphotypes could sweep, replace one another, or coexist (Fig. 5 and Supplementary Data 1). Sequencing of individual clones and population-level sequencing confirmed the genetic basis and frequencies of observed morphotypes. Hypermucoid *wzc*-mutant morphotypes appeared in all strains and typically dominated the populations of C3091 and DA14734 (Fig. 5a and b). In one C3091 lineage, hypermucoid and wrinkly morphotypes with equally high biofilm capacity coexisted at stable (but different) frequencies throughout all six cycles (Fig. 5a). When these clones were mixed at equal ratios together with the parental strain and subjected to a head-to-head competition on pegs, the ratios stabilized at the same levels observed during evolution, suggesting a possible cooperative interaction (e.g., due to niche construction) (Fig. 5a). However, the reverse balance occurred when the same mutants were competed in shaking planktonic conditions. For IA565, hypermucoid morphotypes showed more dynamic trajectories (Fig. 5c-f). They were sometimes replaced by the translucent (non-capsulated) morphotype (Fig. 5e) or clones with parental-like colony morphologies (Fig. 5d, f). However, this clonal interference depended on the composition of the evolving population and possibly the surface. For example, the same hypermucoid mutant (Wzc$^{G538A}$+Prc$^{I465N}$) dominated on silicone + fibrinogen (Fig. 5c) but was outcompeted by a translucent clone on silicone (Fig. 5e). Head-to-head competitions with evolved clones from these lineages recapitulated the same selection dynamics on pegs (Fig. 5c, e). Overall, the evolutionary dynamics revealed rapid population sweeps, strong genetic epistasis, and surface-dependent selection pressures shaping biofilm capacity and morphotype diversity.

## Various levels of mucoviscosity enhance biofilm formation but yield distinct phenotypes across strains

Wzc mutations were shared across all strain backgrounds and targeted conserved residues in the periplasmic and cytoplasmic regions, including the kinase domain (Fig. 6a). A constructed in-frame deletion of *wzc* or selected mutants with *wzc* frameshifts or IS-insertions resulted in non-mucoid colonies. This confirms that the selected missense mutations do not result in loss of function, but instead alter Wzc activity. To quantify mucoviscosity among mutants, we used the sedimentation resistance assay[44]. Wzc missense mutants showed a marked increase in mucoviscosity (Fig. 6b) consistent with their biofilm phenotype on pegs and colony morphotypes (Fig. 1d and video 1 available on figshare (https://doi.org/10.6084/m9.figshare.29816147). However, the phenotypic properties conferred by these mutations varied by strain background, possibly due to differences in capsule types. For example, Wzc mutants in C3091 and DA14734 were nearly impossible to pellet by low-speed centrifugation, while IA565 were less hypermucoid in liquid culture (Fig. 6b), and formed less biomass on the pegs even when mutations occurred at identical amino acid positions (Fig. 6c). Position P642 (643 in IA565) in the kinase domain was targeted by mutations in all three strain backgrounds. All these mutants formed increased biomass on silicone, while the particular amino acid change gave marked differences in biomass on silicone with fibrinogen (Fig. 6c). Consecutive loss of capsule in *wzc*-mutants by accumulation of additional inactivating mutations could result in opposite effects depending on the mutated target (Fig. 6d). Inactivating mutations in *wzc* generally led to a reduction in biomass on all surfaces while loss of capsule by inactivation of *wzb* increased biomass on silicone.

Scanning electron microscopy (SEM) of the most hypermucoid *wzc*-mutants showed a more continuous biofilm architecture on the surface (Fig. 6e). Loss of capsule in *wzc*-mutants by inactivation of *wzb* instead led to cells aggregating into discrete clusters on the surface, indicating a shift from surface interaction to cell–cell aggregation (Fig. 6e, right panels). Transmission electron microscopy (TEM) of DA14734 Wzc$^{P642S}$ showed an increased capsule thickness, cell envelope distortions, less defined LPS, and apparent capsule shedding from

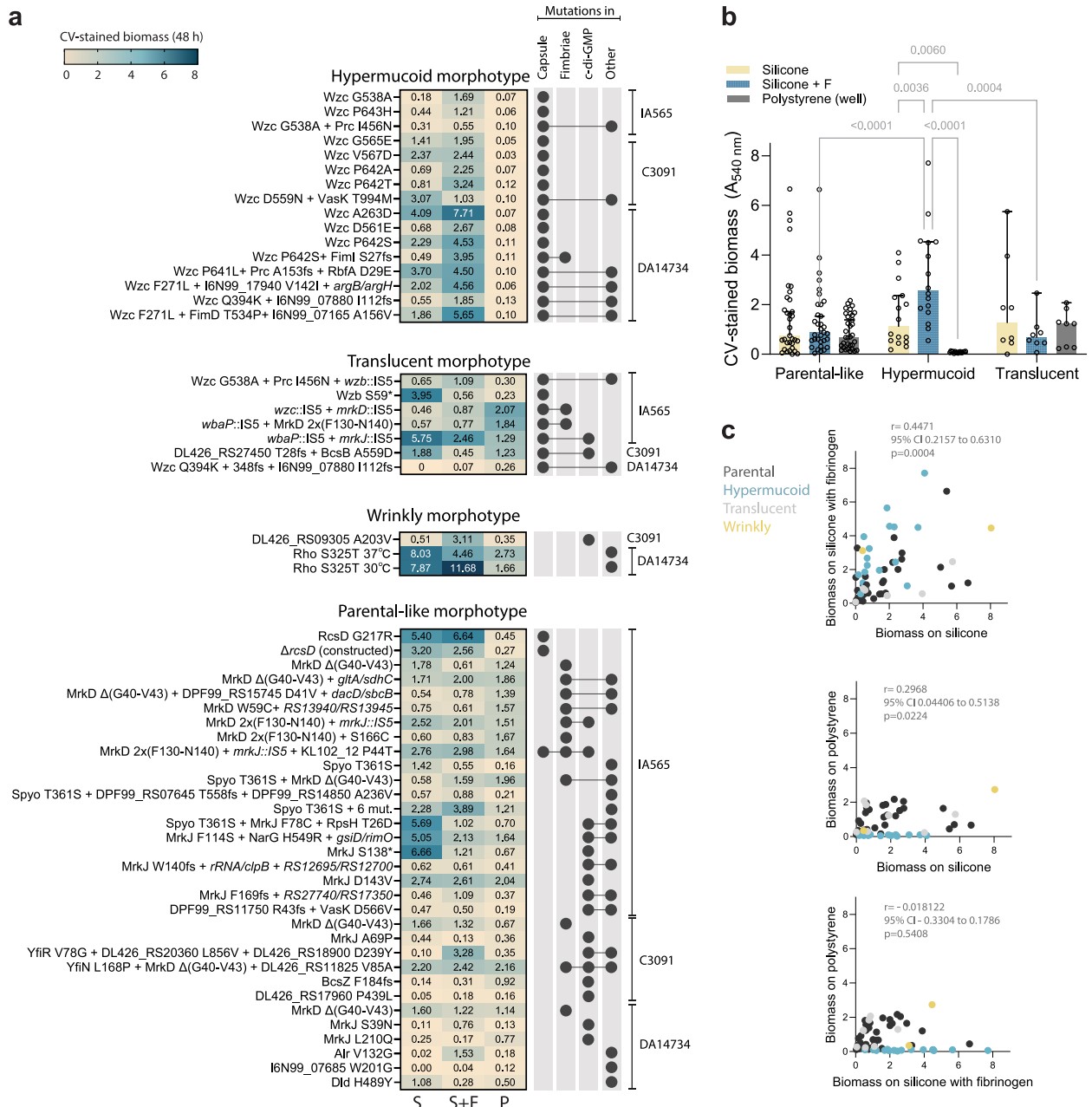

**Fig. 4 | Biofilm capacity by different morphotypes. a** Genetic basis and biofilm capacity of individual mutants displaying specific morphotypes. Heatmap values show the means of 4 biological replicates. Combinations of mutational target functional groups are shown to the right. **b** Biofilm capacity of sequenced clones with unique mutations and different colony morphologies from all strain backgrounds (parental-like: $n = 33$, hypermucoid: $n = 16$, translucent: $n = 8$) on silicone-coated pegs (yellow), silicone pegs with additional fibrinogen coating (light blue), and in polystyrene microtiter plates (gray). Biofilms were grown for 48 h and stained using crystal violet (CV). Bars show means with 95% CI, and each dot represents the mean of 4 biological replicates per clone. The effect of morphotype and biofilm formation surface was assessed using two-way ANOVA; two-sided $p$ values exceeding significance are shown. **c** Interconnection between biofilm formation on different surfaces for individual clones ($n = 59$). Parental morphotypes are in dark gray, hypermucoid in light blue, translucent in gray, and wrinkly in yellow. Pearson correlation coefficients (r) with 95% CI and two-sided $p$ values are shown. Source data are provided as a Source Data file.

the cell surface (Fig. 6f). We also observed increased electron density in the periplasm, which could suggest accumulation of partly- or differently synthesized CPS components, as previously observed for mutations that block specific steps in capsule biosynthesis[45]. Polysaccharide extracts from Wzc mutants were highly viscous following ethanol precipitation, in contrast to the compact pellets from the parental strains, indicating the presence of more and possibly longer polysaccharide molecules in the mutants. Similar findings have been reported for Wzc missense mutants in *A. baumannii*, *A. venetianus*[46,47], and recently in UTI isolates of *K. pneumoniae*[48]. Polysaccharide content

could not reliably be measured by uronic acid assay for the most hypermucoid mutants in the C3091 and DA14734 backgrounds, but all *wzc* mutants in the IA565 strain background showed increased non-attached polysaccharides (Supplementary Fig. 4). Overall, our findings suggest that the hypermucoid morphotype mediated by Wzc mutations is not solely due to capsule overproduction. Instead, it reflects an altered capsule architecture and export that translates into biofilm phenotypes differing from other hypermucoid variants, such as those caused by mutations in the capsule regulator YrfF, which we previously identified in a wound isolate[22].

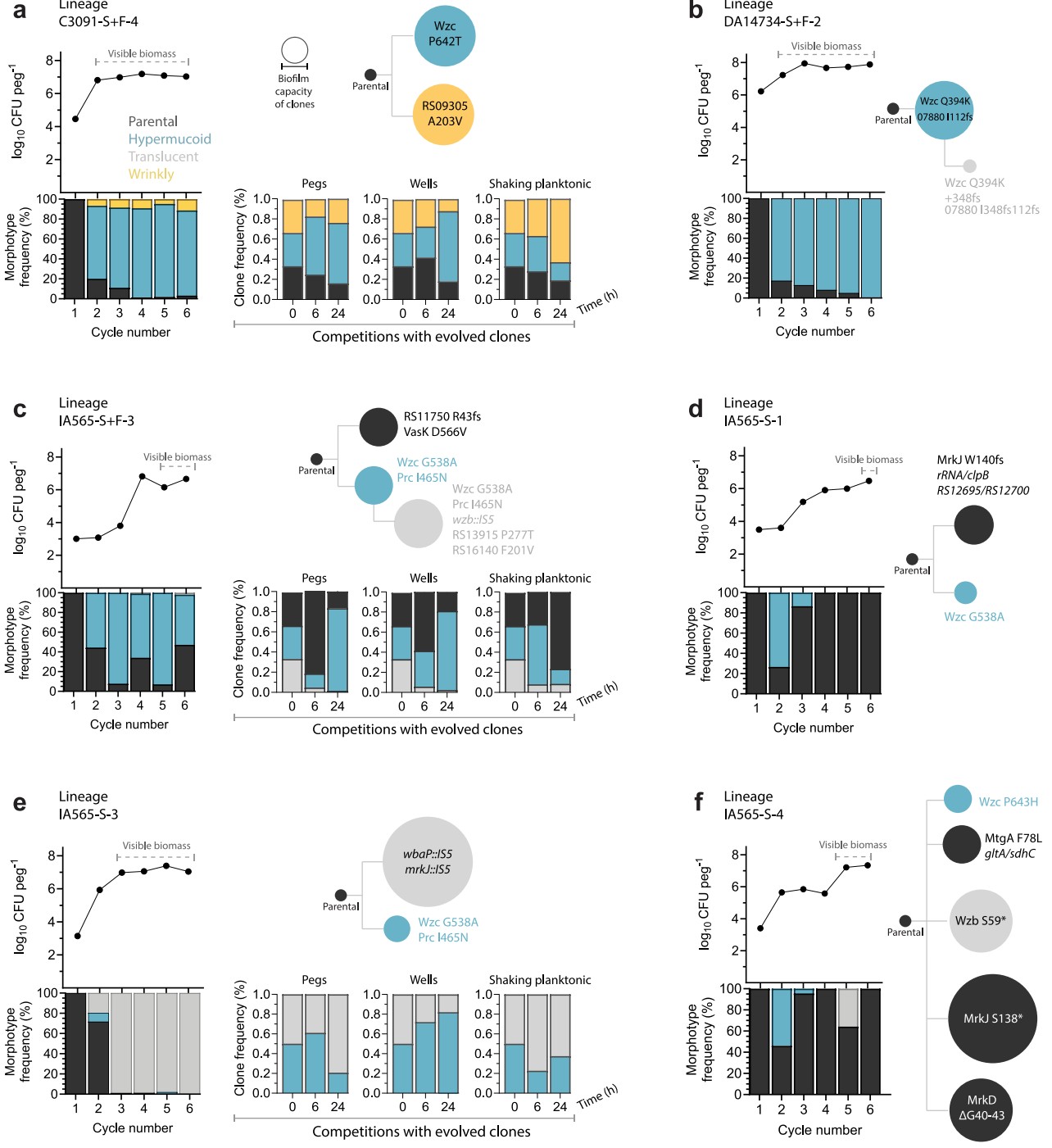

**Fig. 5 | Examples of evolutionary dynamics and characteristics of clones and populations.** Panels **a**–**e** and **f** display different lineages and consist of an upper graph illustrating the biofilm population size during cycling (timepoints where the biomass on the peg became visible by the naked eye is indicated), a lower bar graph with the frequencies of different morphotypes in the evolving population based on the colony count of the biofilm disrupted from the peg, and a cladogram depicting the relatedness of mutants with the size of the nodes indicating their respective relative biofilm capacity (CV-stained biomass at 48 h on the same peg surface as during cycling, means of $n = 4$). Panels **a**, **c**, and **e** also display 24 h competitions (means of $n = 3$) between isolated clones on pegs, in the wells during biofilm growth, and during shaking planktonic growth in tubes. Parental morphotypes are in black, hypermucoid in light blue, translucent in gray, and wrinkly in yellow. The full set of evolutionary dynamics and morphotype frequencies for all evolved lineages is illustrated in Supplementary Data 2. Source data are provided as a Source Data file.

## Structural variations in the lectin domain of MrkD modulate initial attachment, biofilm architecture, and surface specificity

Type 3 fimbriae are key mediators of *K. pneumoniae* attachment to catheter-associated surfaces[9,49]. They consist of the outer membrane usher protein MrkC, the major fimbrial subunit MrkA, the putative stability protein MrkF, and the tip-adhesin MrkD (Fig. 7a). MrkD is

predicted to have a two-domain structure with a lectin domain that mediates receptor binding and can adopt a high-affinity conformation (amino acid residues 24–184), and a pilin domain (amino acid residues 185–332) anchoring the adhesin to the fimbrial shaft[11]. All our selected MrkD mutations mapped to the lectin domain (Fig. 7a). The mutation spectrum was most diverse in IA565, where multiple *mrkD* alleles

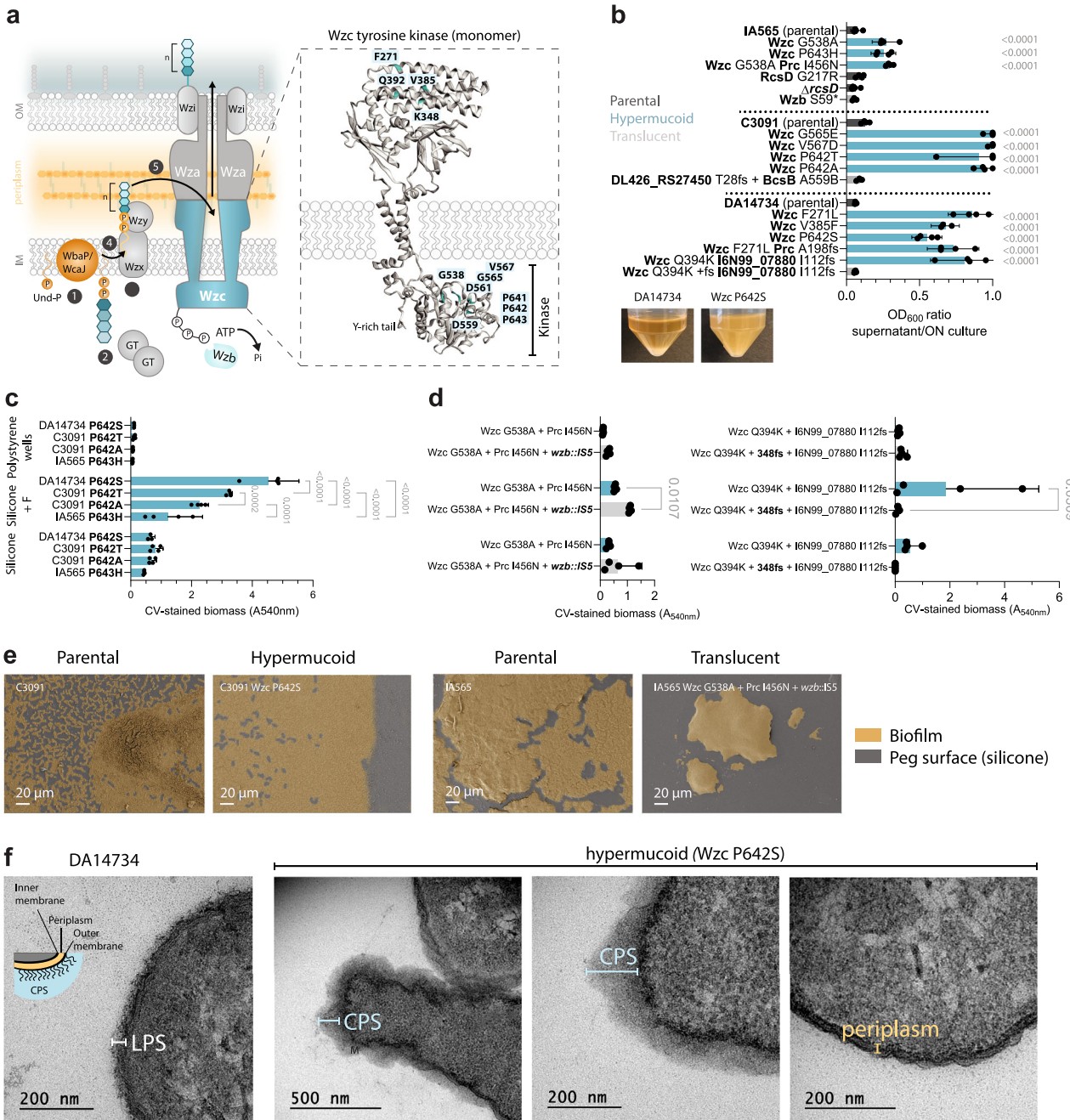

**Fig. 6 | Characteristics of capsule mutants. a** Schematic representation of capsule synthesis in *K. pneumoniae* focusing on the Wzc protein and mutations selected during experimental evolution. The Wzc structure is a monomer (chain A) from *E. coli* (PDB 7NHR). **b** Sedimentation assay showing the ratio between optical density (600 nm) of the supernatant after versus before low-speed centrifugation. Bars represent the means of 4 biological replicates with 95% CI. Blue bars show hypermucoid *wzc* mutants, dark gray – parental-like colony morphology, and light gray – non-mucoid (translucent). Images below show parental DA14734 (left) and *wzc*^P642S (right) cultures after low-speed centrifugation at 1000 × *g* for 5 min. Each mutant was compared against the respective parental strain using one-way ANOVA, followed by Dunnett's T3 multiple comparison test; two-sided *p* values exceeding significance (<0.05) are shown. **c** CV-stained biofilm biomass of Wzc mutants in the P642/643 position from different strain backgrounds on silicone pegs (S), silicone + fibrinogen pegs (S + F), and in polystyrene wells (P) after 48 h. Data show

means of 4 biological replicates with 95% CI. The effects of strain and surface were assessed using two-way ANOVA, followed by Tukey's multiple comparison test to compare mutants on each surface; two-sided *p* values are shown. **d** CV-stained biofilm biomass of hypermucoid mutants and translucent ones derived from them in different strain backgrounds. Data show means of 4 biological replicates with 95% CI. The effects of mutation and surface were assessed using two-way ANOVA, followed by Šídák's multiple comparison test to compare mutants on each surface; two-sided *p* values are shown. **e** SEM images of 48 h biofilms formed by capsule mutants on silicone-coated pegs. Raw images have been color-modified to distinguish biofilm-covered areas. Representative images from three biological replicates are shown. **f** Transmission electron micrographs of the DA14734 parental strain and the *wzc*^P642S mutant. Representative images from three biological replicates are shown. Source data are provided as a Source Data file.

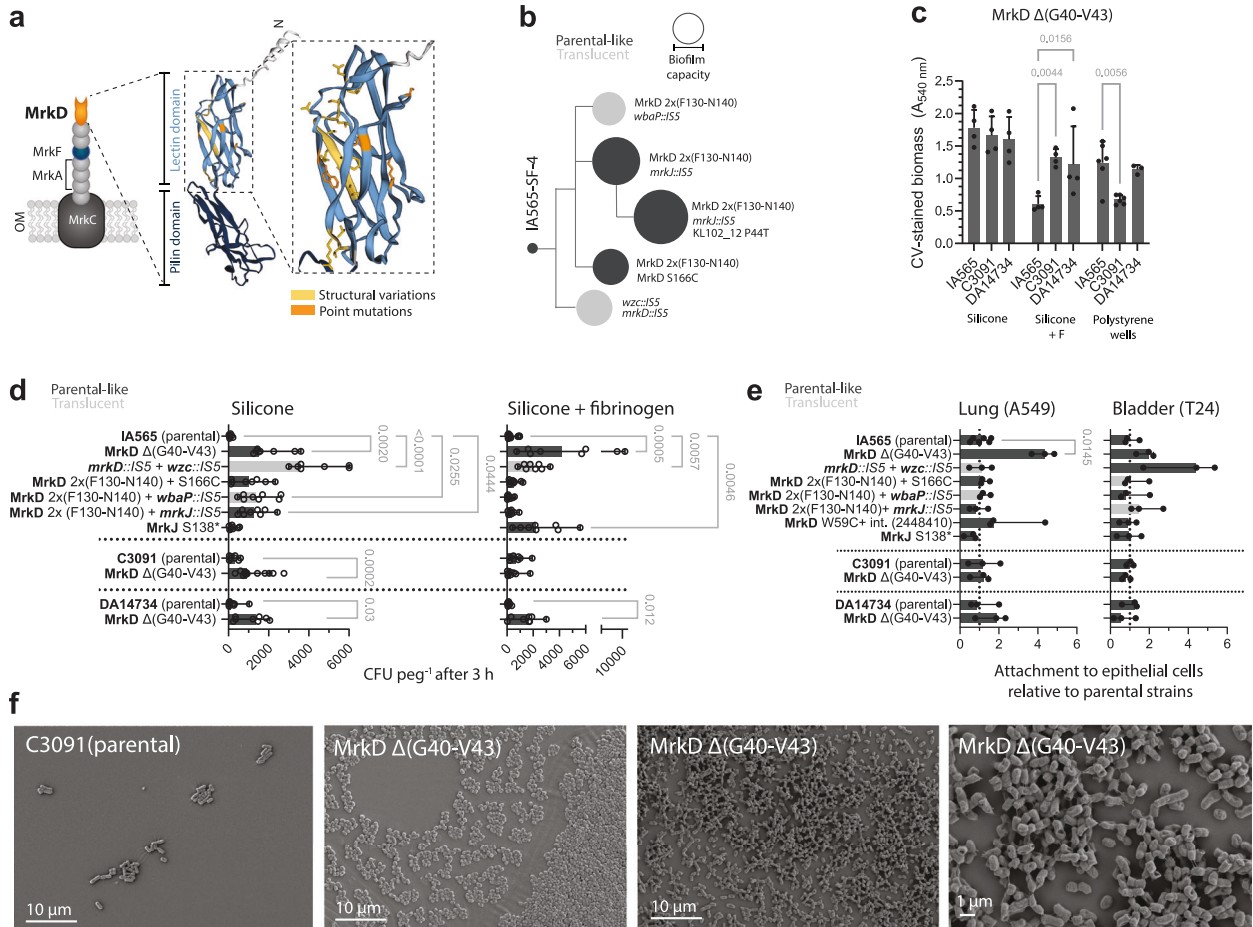

**Fig. 7 | Characteristics of selected structural type 3 fimbriae mutants.**
**a** Schematic illustration of type 3 fimbriae structural components with a focus on the MrkD tip adhesin. Predicted MrkD protein structure (AF-P21648-F1-v6) with marked mutated positions shown. **b** Evolutionary trajectories in the IA565-SF-4 lineage. Cladogram depicting the relatedness of mutants with the size of the nodes indicating their respective relative biofilm capacity (CV-stained biomass at 48 h) of four biological replicates for each mutant on silicone with fibrinogen (the same conditions as used during evolution). Dark gray – parental-like colony morphology, and light gray – non-mucoid (translucent). **c** CV-stained biofilm biomass of MrkD Δ G40-V43 mutants from different strain backgrounds on silicone pegs (S), silicone + fibrinogen pegs (S + F), and in polystyrene wells (P) after 48 h. Data show means of 4 biological replicates for silicone and silicone + fibrinogen, and 6 biological replicates for polystyrene, with 95% CI. The effect of strain and surface was assessed using a two-way ANOVA, followed by Tukey's multiple comparison test to compare mutants on each surface; two-sided *p* values are shown. **d** Attachment to

pegs after 3 h of growth. Data show the medians from 8 biological replicates with 95% CI. Dark gray–parental-like colony morphology, and light gray–non-mucoid (translucent). Comparisons between parental strains and mutants were made using the Kruskal-Wallis test, followed by Dunn's multiple comparison test for the IA565 background and the Mann-Whitney test for C3091 and DA14734 backgrounds; two-sided *p* values exceeding significance (<0.05) are shown. **e** Attachment of exponentially grown bacteria to A549 lung and T24 bladder epithelial cells. Dark gray – parental-like colony morphology, and light gray–non-mucoid (translucent). Data show medians from 3 independent experiments (with two technical replicates per experiment), with 95% CI. Statistical significance was assessed by one-way ANOVA followed by Dunnett's multiple comparison test; two-sided *p* value exceeding significance (<0.05) is shown. **f** SEM images of parental C3091 (left) and *mrkD* mutants (middle and right) grown for 16 h on silicone-coated pegs. Representative images from three biological replicates are shown. Source data are provided as a Source Data file.

coexisted in the same evolving population or were combined in the same clone (Fig. 7b). Notably, the in-frame deletion MrkD$^{\Delta(G40-V43)}$ was selected in all three strains, and MrkD$^{W59C}$ appeared in both IA565 and C3091. The biofilm capacity of the MrkD$^{\Delta(G40-V43)}$ mutation was the same in all three strain backgrounds on silicone but showed a strain dependency on silicone with fibrinogen and on polystyrene (Fig. 7c).

Since fimbrial adhesins mediate the initial attachment of *K. pneumoniae* to abiotic surfaces[11,50,51], we assessed attachment after three hours of incubation to pegs with or without fibrinogen (Fig. 7d). Mutants showed a 10 to 50-fold increase in early attachment to silicone compared to the parental strains. Interestingly, an IA565 capsule-loss mutant with an IS5 insertion in *mrkD*, which eliminates the tip adhesin, was particularly proficient in early attachment to silicone pegs. Although such a change might seem counterintuitive, MrkD is dispensable for biofilm formation on abiotic surfaces as long as MrkA is still present[52]. Mutants with a ten-amino-acid duplication (F130-N140)

in *mrkD* showed enhanced binding exclusively to silicone. MrkD$^{\Delta(G40-V43)}$ also improved attachment to silicone in all strains, with the most substantial effect in IA565, especially in the presence of fibrinogen. SEM imaging revealed that MrkD$^{\Delta(G40-V43)}$ mutants formed discrete cell clusters rather than a continuous monolayer at the early stages (16 h) of biofilm formation (Fig. 7f), reminiscent of the architecture seen in capsule-loss mutants (Fig. 5d).

To evaluate the mutational effects on interactions with biological surfaces, we assessed attachment to lung and bladder epithelial cells (Fig. 7e). MrkD$^{\Delta(G40-V43)}$ significantly increased attachment to lung epithelial cells within 30 minutes, but only in the respiratory isolate IA565. The same mutation in C3091 or DA14734, which carry a different *mrkD* allele, did not change the attachment to lung or bladder epithelial cells, suggesting a strain-specific behavior. MrkD$^{W59C}$ showed a trend towards increased lung cell attachment, while the capsule-deficient *mrkD::IS5* mutant attached better to bladder cells. Thus, the strong

biofilm-enhancing MrkD variants confer highly specific phenotypes, with their functional impact shaped by strain background, surface type, and host cell context. Their evolutionary fate is likely to differ across niches with different abiotic and biological surfaces.

## Regulation of type 3 fimbriae and cellulose production is inter-connected in c-di-GMP-related mutants

The signaling molecule c-di-GMP regulates numerous processes, including the production of fimbrial adhesins and polysaccharides in many bacterial species[53–55]. Regulation of the type 3 fimbriae *mrk* operon is the most well-characterized example in *K. pneumoniae*[40,56–58]. The activity of the transcriptional regulator MrkH depends on the intracellular c-di-GMP levels, which are modulated by a network of phosphodiesterases (EAL domain) and diguanylate cyclases (GGDEF domain) with opposing activities, including the well-characterized MrkJ phosphodiesterase and the YfiN diguanylate cyclase that is controlled by the repressor protein YfiR (Fig. 8a). Seemingly inactivating mutations in *mrkJ* were frequently selected in IA565, while mutations in *yfiN* and *yfiR* were found in C3091 (Fig. 3b). Three additional uncharacterized proteins with EAL and/or GGDEF-domains were also targeted (Fig. 8a), including the DL426_RS09305$^{A203V}$ mutation in a membrane sensor domain conferring the wrinkly phenotype in C3091. To assess their functional roles, we constructed in-frame deletions of these putative c-di-GMP-related genes and *mrkJ* and subjected them to the same assays as the evolved mutants. Although not directly connected to c-di-GMP, we also assayed the Rho$^{S325T}$ mutant with the temperature-dependent wrinkly morphotype since it is likely associated with transcriptional changes.

We hypothesized that the increased biofilm capacity in these mutants would correlate with overproduction of type 3 fimbriae, via higher c-di-GMP levels. Indeed, most mutations led to substantial upregulation of *mrkA*, with some resulting in increases exceeding 100-fold. However, the expression was mutation-, strain-, and growth state-dependent (Fig. 8b). In comparison, the within-host evolved isolates from the outbreak[22] with mutations in c-di-GMP-related genes exhibited an even stronger (up to >300-fold) increase in *mrkA* expression, particularly during the stationary phase. Expression of *mrkA* (Fig. 8b) and biofilm phenotype (Supplementary Fig. 5) in the constructed deletion mutants confirmed that selected mutations in DPF99_RS11750 and DL426_RS17960 are loss-of-function mutations. In contrast, deletion of DL426_RS09305 did not affect biofilm, morphotype, or *mrkA* expression, suggesting that the A203V mutation alters protein function rather than abolishing it. It is clear that differences in *mrkA* expression alone cannot fully explain the biofilm capacity in *mrkJ*, *yfiN/R*, and other EAL/GGDEF protein mutants (Supplementary Fig. 6), illustrating that a combination of factors contributes to the phenotype. SEM imaging of the IA565 MrkJ$^{S138*}$-mutant, which exhibited the strongest biofilm phenotype, revealed an initial biofilm topology resembling that of loss-of-capsule mutants, with 80–100 μm cell clusters, whereas at later time points it developed multilayer biofilms with prominent ECM (Fig. 8d).

In addition to type 3 fimbrial expression, YfiN/R has been connected to the expression of the *bcs* cellulose operon[58]. The selection also resulted in changes in *bcsB*, which encodes a co-catalytic membrane protein involved in the c-di-GMP-dependent synthesis and transport of cellulose, and the BcsZ periplasmic endoglucanase, which hydrolyzes cellulose. Given that cellulose regulation in *K. pneumoniae* biofilms is poorly understood, we determined both the mRNA levels of *bcsA* and the effect of cellulase (1,4-endoglucanase) on growing biofilms. While *bcsA* mRNA levels were unchanged in most mutants (Fig. 8b), biofilms formed by certain EAL/GGDEF (MrkJ$^{S138*}$, MrkJ$^{D143V}$) and cellulose synthesis mutants (BcsB$^{A559D}$) were sensitive to cellulase treatment (Fig. 8e), suggesting cellulose as a key ECM component. In other cases, the reduction in biomass was more moderate, and it was more pronounced in the parental C3091 than in IA565 (Supplementary

Fig. 7). In contrast, biofilms of hypermucoid Wzc mutants were unaffected by cellulase, indicating a different biofilm composition where other ECM components likely dominate or protect the cellulose from degradation (Fig. 8e and Supplementary Fig. 5). These results suggest that cellulose contributes significantly to the biofilm structure in specific evolved mutants, and that the c-di-GMP dependent regulation is not transcriptionally mediated, in contrast to type 3 fimbriae. This observation is in agreement with a reported direct interaction between c-di-GMP and the PilZ domain of BcsA in other bacteria[53,59,60]. Despite displaying changes in c-di-GMP-dependent processes, most mutants did not exhibit increased global c-di-GMP levels by LC-MS (Fig. 8f), suggesting that local or transient concentration changes are the primary drivers in the c-di-GMP-regulated pathways in these mutants, or more subtle changes that are not visible from bulk ex vivo measurements.

The Rho$^{S325T}$ mutant with exceptionally robust biofilm formation (Fig. 4a) was among the mutants with the highest increase in both *mrkA* and *bcsA* expression (Fig. 8b). The biofilm was also sensitive to cellulase (Fig. 8e), indicating a strong role for cellulose in its ECM. However, unlike *mrkJ* mutants, Rho$^{S325T}$ upregulated *bcsA* transcription by more than 80-fold, suggesting an alternative regulatory pathway that is probably not directly linked to c-di-GMP. In summary, our findings reveal that biofilm-enhancing mutations in *K. pneumoniae* frequently target c-di-GMP-related pathways, but their effects are multifaceted and context-dependent. Type 3 fimbriae and cellulose production are tightly interconnected yet regulated through distinct mechanisms–transcriptional for fimbriae and posttranslational for cellulose. These insights underscore the complexity of biofilm regulation and highlight the diverse evolutionary strategies *K. pneumoniae* can employ to adapt to surface-associated growth.

## Acute virulence can be lost as a trade-off for increased biofilm capacity

Although the *K. pneumoniae* strains were cycled without any host material (except for fibrinogen) or immune pressure, many of the selected mutations affected structures and pathways known to be critical for pathogenesis. The complement system is one of the primary defenses against pathogens in the human host, and serum resistance is a well-established predictor of systemic infection potential[61]. As expected, loss-of-capsule mutants were sensitive to human serum; however, despite their hypermucoid phenotype, *wzc* mutants also exhibited extreme serum sensitivity with >5 log increased killing within three hours (Fig. 9a). This phenotype was consistent across hypermucoid *wzc* mutants from both the experimental evolution and the clinical outbreak[22]. Additional mutations in *prc*, which is important in complement evasion in *E. coli*[62], did not rescue this phenotype. However, the sensitivity was abolished when the mutants were grown as a biofilm prior to serum exposure (Fig. 9b). In contrast, evolved mutational changes in *rcsD* or the constructed knock-out did not affect serum sensitivity, consistent with their parental-like morphotype. Interestingly, in C3091, MrkD$^{Δ(G40-V43)}$ and MrkJ$^{A69P}$ mutants showed better survival in human serum, indicating that fimbrial adhesins and their regulation may contribute to complement resistance. This highlights the importance of cell surface structures beyond LPS and capsule in modulating complement activation or deposition in a strain-dependent manner[63,64].

To assess virulence in a more complex host-like environment, we tested the infection potential of a set of mutants affecting capsule, fimbriae, and c-di-GMP signaling in *Galleria mellonella* larvae (Fig. 9c–e). This infection model, with high similarity to the mammalian innate immune system, has become the preferred model for multiple opportunistic pathogens, including classic pathotype *K. pneumoniae*, which performs poorly in murine models[65–67]. Virulence outcomes varied by strain and mutation. For example, *wzc* point mutations at amino acid position P642/643, which conferred high

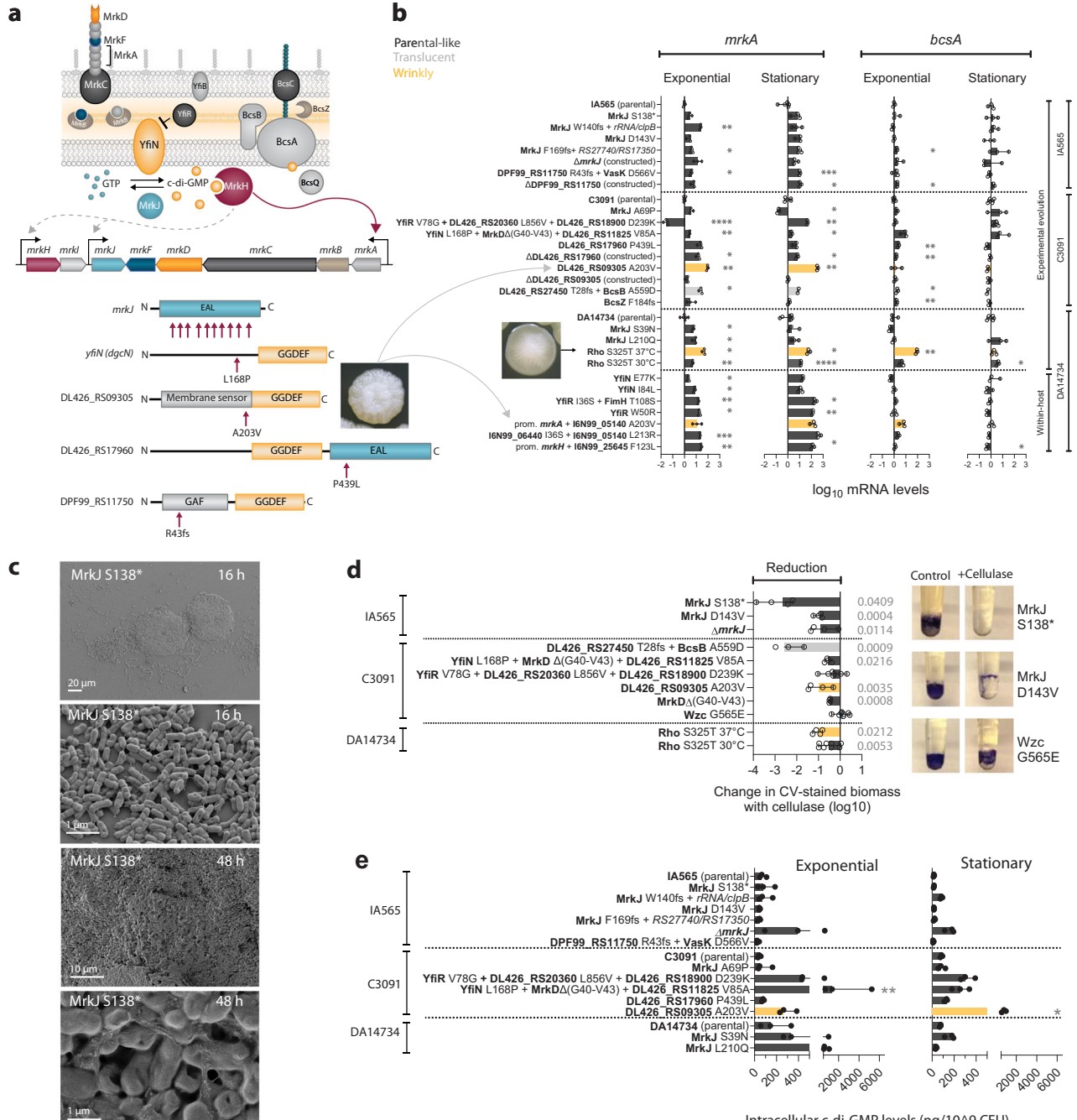

**Fig. 8 | Interconnection between type 3 fimbriae, cellulose, and c-di-GMP in biofilm- and within-host evolved mutants. a** Schematic illustration of the type 3 fimbriae operon organization and expression regulation. Protein domain structures based on protein BLAST results in uncharacterized EAL/GGDEF proteins from IA565 (DPF99_RS11750) and C3091 (DL426_RS17960 and DL426_RS09305). Arrows mark the mutations. A203V in DL426_RS09305 results in a unique, wrinkly colony morphology, which was also observed in a clinical UTI isolate with the same mutation[22]. **b** *mrkA*, and *bcsA* mRNA levels in selected mutants during exponential and early stationary growth, normalized to the respective parental strain. Parental morphotypes are in dark gray, translucent in gray, and wrinkly in yellow. Data show means from three biological replicates with 95% CI. Comparison between parental strains and mutants was done using one-way ANOVA, followed by an unpaired t-test with Welch's correction; two-sided *p* values exceeding significance ($p < 0.05$) are shown $*p < 0.05$, $**p < 0.01$, $***p < 0.001$, $****p < 0.0001$. Mutations shown for within-host evolved isolates (DA14734) at the bottom refer to mutations most likely

responsible for the biofilm phenotype; however, the isolates contain additional changes[22]. **c** SEM images of *mrkf*[S138*] (IA565) on silicone pegs after 16 h and 48 h of growth. **d** Effect of cellulase treatment on biofilms growing on silicone-coated pegs for 48 h. Parental morphotypes are in dark gray, translucent in gray, and wrinkly in yellow. Data show means of 8 biological replicates with 95% CI. Statistical significance was assessed by a paired Student's t test between control and treated biofilms for each mutant; two-sided *p* values exceeding significance (<0.05) are shown. Images on the right show the CV-stained biofilms on pegs grown with and without cellulase. **e** Quantification of c-di-GMP by LC-MS in exponential and early stationary cultures. Parental morphotypes are in dark gray, translucent in gray, and wrinkly in yellow. Data are presented as means of 3 biological replicates with 95% CI. Comparison between parental strain and mutants was done using the Kruskal-Wallis test, followed by Dunn's multiple comparison test. Differences are considered statistically significant when $p < 0.05$. Source data are provided as a Source Data file.

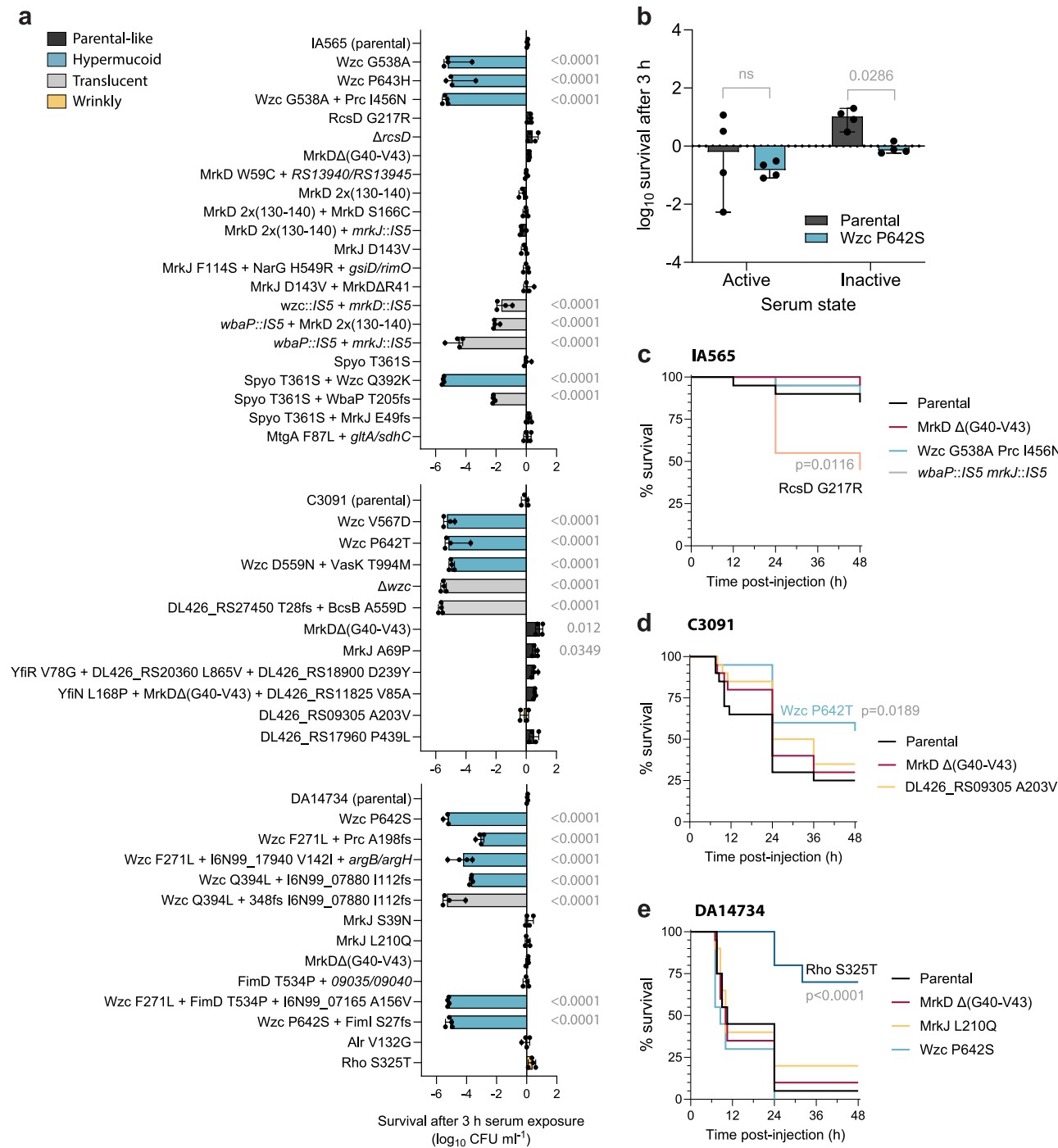

**Fig. 9 | Virulence features in evolved mutants. a** Serum killing assays with planktonic cultures expressed as the $\log_{10}$ difference in CFU/ml after 3 h exposure to human serum. Parental morphotypes are in dark gray, hypermucoid in light blue, translucent in gray, and wrinkly in yellow. Bars show medians of $n = 4$ biological replicates with 95% CI. Comparisons to parental strains were performed using one-way ANOVA followed by Dunnett's T3 multiple comparison test; two-sided $p$ values exceeding significance ($p < 0.05$) are shown. **b** Serum killing assay with biofilms of the parental strain DA14734 and the Wzc$^{P642S}$ mutant. The biofilms were grown for 48 h on silicone-coated pegs and then exposed to human serum for 3 h. Parental is in dark gray, hypermucoid Wzc P642S in light blue. Bars show medians of $n = 4$ biological replicates with 95% CI. Survival of parental strain and the Wzc mutant was compared using the Mann-Whitney test; two-sided $p$ value is shown, ns not significant. Survival of *G. mellonella* larvae infected with mutants ($10^5$ CFU) derived from **c** IA565, **d** C3091, and **e** DA14734 strains. Parental morphotypes are in black, *mrkD* in red, hypermucoid in light blue, translucent in gray, and wrinkly in yellow. Survival curves ($n = 20$ larvae, 10 larvae for one biological replicate) were compared by the log-rank (Mantel-Cox) test; $p$ values for significantly different comparisons are shown ($p < 0.05$). Source data are provided as a Source Data file.

serum sensitivity in all strains, significantly decreased larval killing in C3091 but not in IA565 or DA14734. For DA14734 Wzc[P642S], there was a tendency towards faster killing, in contrast to other *wzc* mutants from the outbreak[22]. The strongly biofilm-forming IA565 RcsD[G217R] had increased larval killing, consistent with its serum resistance. In contrast, the even better biofilm-forming mutant DA14734 Rho[S325T] was partially attenuated in larvae, indicating that biofilm capacity alone does not necessarily correlate with increased killing. Notably, no trade-offs in planktonic growth rates were seen for the evolved mutants (Supplementary Fig. 8). These results show that mutations enhancing biofilm formation can lead to significant trade-offs in acute virulence, particularly in serum resistance and host killing. The decoupling of biofilm capacity from virulence underscores the complexity of *K. pneumoniae* pathoadaptation and suggests that biofilm-promoting mutations may favor persistence over acute infection in specific niches.

## Discussion

Since interaction with surfaces is often a central part of nosocomial *K. pneumoniae* infections[6,7,68,69], exploring the adaptive changes that lead to altered biofilm phenotypes can provide essential insights into the pathoadaptivity of this bacterium. Here, we demonstrate that *K. pneumoniae* can rapidly evolve improved biofilm formation when subjected to cycling in vitro on abiotic surfaces, including those mimicking the typical catheter surface. In a clinical setting, this could translate into the selection of bacterial subpopulations that initiate infections in intubated or catheterized patients due to altered adhesion abilities. By comparing the in vitro evolution of the outbreak clone DA14734 to changes identified in patient isolates[22], we demonstrate that short-term experimental evolution can replicate some of the most common adaptations selected within the host and possibly explain why they were established during infection. The morphotypes we identify as advantageous in biofilms, for example, hypermucoid and translucent, have also been linked to globally transmitted multi-resistant high-risk cKp[35]. A rarely reported wrinkly morphotype was recently connected to adaptation in carbapenem-resistant hypervirulent *K. pneumoniae*[70] and is generally associated with chronic infections in other bacteria[15,32,36,55].

The evolutionary trajectories of the three opportunistic strains isolated from UTI, wound, and trachea exhibited substantial genetic parallelism. While we observe strong convergent evolution across strains, there appears to be a strain-specific bias in the changes that are most frequently selected and the strength of their phenotypic effects. This illustrates that while certain shared factors are essential for biofilm formation in *K. pneumoniae*, their contributions can vary depending on the genetic background, possibly diversifying the adaptability to specific niches. Strain-dependent selection bias was apparent at the target level (capsule vs. fimbrial mutations), at the gene level (e.g., *mrkJ* vs. *yfiN/yfiR*), and in the biofilm phenotype (*wzc* mutants in IA565 vs. C3091/DA14734). Missense mutations in *wzc*, leading to a hypermucoid phenotype, were more frequently selected in C3091 and DA14734, and the phenotype was significantly more pronounced in these strains. Differences in overall capsule-mediated phenotypes are likely due to variations in capsule types, and strain influences on capsule involvement in *K. pneumoniae* biofilms have also been observed by others[71]. Similarly, mutations affecting the regulation of type 3 fimbriae and the structural component MrkD were often co-selected with mutations that abolished capsule production, a finding similar to a recent study[72]. Such mutants likely expose fimbriae more and were primarily found in the IA565 background. This could indicate a higher evolvability of the *mrkD*-allele in IA565, which differs from the most abundant and stronger binding allele present in the other two parental strains[11]. IA565 also carries an additional *mrkD_{IP}* on a plasmid[73], although we found no mutations in that locus. Differences in specific c-di-GMP-related changes, such as different levels of fimbrial

expression or cellulose production, are likely attributed to additional networks of EAL/GGDEF domain proteins in specific strains that drive particular responses, as c-di-GMP signaling networks show extensive variation even within species[74–76]. Considering the vast genetic diversity of *K. pneumoniae*[8], it is essential to study such strain dependencies systematically and identify the key factors influencing the response in different genetic backgrounds.

Our results suggest that biofilm growth serves as a selective pressure for the emergence of both the hypermucoid and capsule-loss mutants, which we also identified as within-host-selected infection site adaptations during the Uppsala hospital outbreak[22]. In parallel with our work, another study reported the selection of *wzc* mutants in "structured environments", which was achieved by serial passaging in statically incubated 24-well plates[77]. How this form of growth relates to cell-cell interactions or to interactions with surfaces was not determined. Widespread alteration or loss of capsule production via *wzc* and *wbaP* mutations has also recently been reported among globally disseminated *K. pneumoniae* ST258 isolates[35]. However, this study suggested that *wbaP* mutants in ST258 form surface-attached biofilms, whereas *wzc* mutants do not[35]. Importantly, we find that *wzc* mutants firmly attach to silicone or fibrinogen, but not to polystyrene microtiter plates, which was the method used by Ernst et al. to measure biofilm formation[35]. On the other hand, we observe apparent phenotypic differences among these mutants across different strain backgrounds, which could account for the discrepancies between the studies. Additionally, in contrast to mutants carrying only *wzc* mutations, within-host-selected isolates with additional genetic changes did not form biofilm under the same conditions[22]. This could be due to additional mutations in other genes that may compensate for specific trade-offs associated with the *wzc* phenotype, although this remains to be determined. Biofilm growth rescues *wzc* mutants from the extreme planktonic serum sensitivity that we otherwise observe. Similarly, LPS mutations that enhance biofilm growth in *P. aeruginosa* can render planktonically growing cells more susceptible to antimicrobial agents[78]. Such trade-offs between planktonic and biofilm growth, which affect survival in the host, could generate subsequent selection for compensatory mutations.

The growing number of studies identifying Wzc as a mutational target linked to infections warrants a deeper understanding of its role in the observed phenotypes. Mechanistically, there are multiple possibilities for how *wzc* mutations could affect the protein function and, consequently, the entire capsule synthesis machinery. For example, alterations in the interaction with the Wza transporter on the periplasmic side could directly affect the export of components from the periplasm. The increased periplasmic density observed with TEM could support this. Alternatively, changed interactions with the Wzc phosphatase in the cytoplasm could interfere with the cycling between the phosphorylated and dephosphorylated states and, thus, the oligomerization of Wzc itself. Altered phosphorylation levels have just been reported in *wzc* mutants from UTI *K. pneumoniae*[48]. The key phenotypic feature we observed in both single *wzc* missense mutants and the combined *wzc* mutants from the outbreak is extreme sensitivity to serum in planktonic conditions. Based on our TEM observations and differences in cell-attached versus free extracellular polysaccharides, we propose that this is due to multifactorial effects on the cell envelope, including increased shedding of high-molecular-weight capsular polysaccharides. To further support this, a weakened association of the capsule with the cell surface in such *K. pneumoniae* mutants has just been reported[48]. In addition, the integrity of the entire cell envelope appears to be compromised, possibly due to the sequestration of undecaprenyl phosphate and the accumulation of partially synthesized components[45]. Frameshift mutations in the Prc protease (processing of PBP3), deletion of which results in leakage of periplasmic proteins in *E. coli*[79], always coincided with *wzc* mutations in our study, raising the question about a possible interplay between

these changes. Inactivation of *prc* via transposon insertion also increases capsule production in *K. pneumoniae*[44,80].

The biofilm-conferring mutations identified here affect factors crucial for *K. pneumoniae* pathogenesis, such as the capsule and type 3 fimbriae, suggesting a potentially altered virulence status. However, virulence profiles in *G. mellonella* larvae did not directly correlate with biofilm capacity, meaning that excellent biofilm-forming strains can have reduced (e.g., *rho*), increased (e.g., *rcsD*), or unchanged acute virulence. Thus, the relationship between biofilm formation and acute/systemic virulence depends on the exact genetic change underlying the biofilm phenotype, which is in agreement with our recent observations in within-host evolved isolates[22]. Similarly, a study of *K. pneumoniae* defective in biofilm formation found no direct correlation between the loss of biofilm formation and the ability to cause lung infections in mice[81]. However, we associated the high biofilm-forming isolates with a gastrointestinal colonization advantage in mice[22]. These results illustrate the complex interplay between genetic variations that mediate the overall phenotype and the within-host selection of bacteria with altered capacities for infection through indwelling medical devices.

## Methods

### Bacterial strains and growth conditions
Three clinical strains of *K. pneumoniae* were used: (i) IA565 (ST105), a clinical tracheal aspirate isolate, originally from the University of Iowa Hospitals and Clinics Special Microbiology Laboratory[27]; (ii) C3091 (ST14), a *K. pneumoniae* UTI isolate from Walter Reed Army Medical Center[26]; and (iii) DA14734 (ST16), an ESBL-producing *K. pneumoniae* that was the index isolate from an outbreak at Uppsala University Hospital[24]. Bacteria were grown in Brain Heart Infusion broth (BHI, Oxoid) both before inoculation and for biofilm growth. For the preparation of electrocompetent cells and serum-killing assays, planktonic bacteria were grown in Lysogeny Broth (Sigma). The composition of sucrose agar used for counter-selection during strain construction: 10 g/L tryptone, 5 g/L yeast extract, 15 g/L agar, 1 mM NaOH, 50 g/L D-sucrose. Antibiotics used to select transformants during strain construction were cefotaxime (10 mg/L, Sigma) and kanamycin (50 mg/L, Sigma).

### Setup for experimental evolution on pegs
The cycling of *K. pneumoniae* IA565, C3091, and DA14734 was performed on a 3D printed FlexiPeg device[23] with high-temperature resin (HT) pegs (Form 2 3D-printer) and silicone (polydimethylsiloxane, PDMS)-coated pegs. Silicone-coated pegs were further coated with 100 mg/L fibrinogen as described before[22] when indicated. Overnight (O/N) cultures of 10 independent lineages for each strain on HT pegs and 6 lineages for each strain on silicone-coated pegs and silicone-coated pegs with additional fibrinogen coating were diluted 100-fold or 10.000-fold in BHI and 150 µl transferred to a 96-well microtiter plate (NuncTM) where the peg lid was then inserted. The biofilms formed on the pegs were transferred to fresh medium after 24 h and were harvested after 48 h by vortexing for 2 min as previously described[23]. Previous experiments have shown that CFUs do not increase substantially after 48 hours in this device; instead, there is an increase in biomass, indicating no further mutation supply. We first performed a pilot cycling on high-temperature resin (HT) pegs as a proof of concept. After each 48-hour cycle, 0.25% or 25% of the disrupted biofilm population was transferred to initiate biofilms on new pegs. No apparent differences were observed in the rate or range of mutations selected with the different bottlenecks, and since the parental strains did not reach as high CFU/peg with the smaller bottleneck, which led to an increased population loss rate, cycling was only continued with the 25% bottleneck in the following experiments. Cycling was continued for 5 cycles (non-coated BHI conditions) or 6 cycles (silicone with/without fibrinogen, BHI).

### Measurement of mutation rates
The spontaneous mutation rates to rifampicin resistance were measured using fluctuation tests in both liquid culture and biofilm. Twenty-four individual cultures for each parental strain were grown in BHI at 37 °C overnight. Liquid cultures were prepared by adding 10,000 cells from each overnight culture to initiate 5 ml cultures in BHI, which were incubated at 37 °C with shaking at 200 rpm. In parallel, a 5 ml biofilm culture in BHI, using a larger peg (15-fold greater surface area for biofilm formation than the FlexiPegs) to accommodate a larger biofilm population, was initiated with the same inoculum and incubated statically at 37 °C. After 24 h of incubation, biofilms were transferred to fresh medium and incubated for an additional 24 h. Before harvest, biofilms were washed three times with PBS and then disrupted by vortexing as described above. The whole harvested biofilm population was pelleted by centrifugation at $8000 \times g$ for 10 min, resuspended in 100 µL PBS, and plated on plates containing rifampicin (Sigma Aldrich) at 100 mg/L. From liquid cultures, 1 ml was pelleted by centrifugation at $8000 \times g$ for 10 min, resuspended in 100 µL PBS, and plated on plates containing rifampicin (Sigma Aldrich) at 100 mg/L. All populations were enumerated by viable counts on LA plates without antibiotics. Mutation rates were calculated using the median method of Lea & Caulson, with 95% confidence intervals, as described in ref. 82.

### Biofilm growth on pegs
The biofilms were grown in the FlexiPeg device for 12, 16, 20, or 48 h according to the general protocol[23]. The biofilms were harvested by vortexing for 2 min or stained with 0.1% crystal violet (CV, Sigma) at different time points as previously described[23].

### Biofilm formation in polystyrene microtiter plates
O/N cultures were diluted 10000-fold in BHI and 150 µl inoculated into a round-bottom 96-well plate (Nunc) to initiate biofilm formation in microtiter plates. The plate was statically incubated for 48 h at 37 °C. After 48 h, unattached cells were removed from the wells, and the plate was washed 3x with 1x sterile PBS, dried for 30 min at 37 °C, and 180 µl of 0.1% CV (Sigma) was added. After 20 min of incubation at room temperature, the plate was extensively washed (3-4x) with PBS to remove any non-bound CV stain, dried for 20 min, and 180 µl of 10% acetic acid was added to solubilize the stain. The level of CV was quantified by measuring the absorbance at 540 nm with a Multiskan™ FC Microplate Photometer (Thermo Scientific). The average value from the blank (acetic acid only) was subtracted from the sample values.

### Biofilm formation in the presence of cellulase
An aqueous solution of cellulase from *Aspergillus niger* (Sigma-Aldrich, product number C2065) was diluted 1:60 in BHI to yield approximately 20 mg/ml, and 150 µl were transferred to a flat-bottom 96-well plate (Nunc). O/N cultures were diluted 1:100 in BHI, and 1.5 µl were transferred to a 96-well plate with BHI + cellulase or BHI only, and the FlexiPeg lid was inserted into the plates. The biofilms were allowed to form for 24 h on silicone-coated pegs, then they were either stained with CV or harvested by vortexing for 2 min and plated on LA plates for CFU counts.

### Competitive growth measurements of mutants
Selected mutants with different morphotypes were competed on the pegs, in 96-well microtiter plates, and in shaking planktonic cultures. Starting O/N cultures (n = 3 biological replicates) grown in BHI were mixed at equal ratios and diluted 10000-fold in BHI to inoculate into a flat-bottom 96-well microtiter plate, where silicone-coated pegs were inserted. Biofilms were allowed to statically form at 37 °C and harvested after 6 h and 24 h. The biofilms were harvested by vortexing as described above and plated for CFU counts on LA plates to enumerate the respective mutants. Populations from the wells, where pegs were

inserted, were also harvested by vortexing and plated for CFU counts. In parallel, the same O/N cultures were mixed and diluted 1000-fold into BHI (total volume of 1 ml) in plastic tubes and grown at 37 °C with shaking (190 rpm). Aliquots for CFU counts were taken after 6 h and 24 h, and dilutions plated for CFU counts.

## Exponential growth rate measurements

O/N cultures were diluted 1000-fold in BHI, and 300 μl were transferred to 100-well honeycomb plates. Four to five biological replicates were used for each isolate. The plate with cultures was incubated in a BioScreen C (Oy Growth Curves Ad Ltd) for 16 h at 37 °C with shaking. $OD_{600}$ measurements were taken every 4 min. Results were analyzed using the R script-based tool BAT 2.0[83] to determine relative growth rates.

## Strain construction by λ Red recombineering

In-frame deletions of genes mutated during the cycling in the biofilm were constructed in IA565 and C3091 background. Strains were first transformed with the pSIM5-CTX and selected on LB agar with cefotaxime (10 mg/L) at 30 °C. pSIM5-CTX carries the temperature-inducible λ Red recombineering system and is derived from the original pSIM5[84]. The *kan-sacB-TO* cassette was amplified by PCR using Phusion High-Fidelity DNA Polymerase (Thermo Fisher Scientific Inc.) with primers including 40 bp flanking homology regions outside the gene of interest. The cultures (50 ml) were grown in LB with 10 mg/L cefotaxime at 30 °C until early exponential phase ($OD_{600}$ - 0.3), and the λ Red system was induced by 15 min incubation in a 42 °C shaking water bath, followed by cooling on ice for 10 min. To make the cells electrocompetent, the cultures were pelleted by centrifugation at 4 °C and washed 4–5 times in sterile 10% glycerol, finally resuspending in 500–800 μl 10% glycerol. Electrocompetent cells (50 μl) were mixed with 100–500 ng of the purified *kan-sacB-TO* cassette with homology regions, transferred to 0.1 cm gap cuvette and electroporated at 2.5 kV. The cells were recovered in 1 ml LB or BHI overnight at 30 °C with shaking. Transformants were selected on LB agar with kanamycin (50 mg/L) at 30 °C and checked for sucrose sensitivity on 5% sucrose plates and the carriage of the pSIM5 plasmid on LB agar with cefotaxime (10 mg/L). Successful transformants were used for another λ Red recombineering step with a linear ssDNA fragment containing 40 bp homologous regions directly upstream and 40 bp downstream of the gene of interest to delete the *kan-sacB* cassette. Transformants were selected on 5% sucrose agar plates and PCR-verified for the correct deletion. Primers used for the construction can be found in Supplementary Data 3.

## Genomic DNA extraction, whole-genome sequencing (WGS), and bioinformatics

To identify the underlying genetic changes that result in better biofilm formation, we whole-genome sequenced selected clones and populations from each lineage. For the pilot HT cycling rounds, we randomly chose one clone per lineage from the endpoint population for whole genome sequencing and further characterization unless we observed different morphotypes, in which case we sequenced one clone of each (Supplementary Data 1). For the lineages evolved on silicone or silicone with fibrinogen, we first screened at least four clones per lineage from the endpoint population for changes in biofilm formation capacity by CV staining (48 h). We sequenced one clone with the highest biofilm-forming capacity from the endpoint populations of each lineage. If there was more than one morphotype showing increased biofilm formation, we sequenced one of each. For lineages, where we observed appearance of morphotypes earlier during cycling or clear changes in population size, we also screened for biofilm formation capacity in the same way as described above, and sequenced clones. In addition, we sequenced populations from selected lineages evolved on silicone or silicone + fibrinogen with dynamic morphotype changes or clear changes in population size during cycling to follow changes in mutant frequencies over time.

Genomic DNA was extracted from 500 μl of O/N cultures using the Epicenter MasterPure™ DNA purification kit (Illumina Inc.) according to the manufacturer's instructions. For population sequencing after evolution, we did a short pre-growth before DNA extraction to reach approximately $4 \times 10^8$ CFU/ml for populations with low CFU (below $10^7$ CFU/peg) numbers. Depending on the lineage, the pre-growth resulted in 5 to 16 generations of growth, and a pilot run showed that this short pre-growth did not change the frequencies of morphotypes in the evolved population. The two parental strains, IA565 and C3091, were sequenced using the Pacific Biosciences II technology at the Science for Life laboratory sequencing facility and assembled to create reference genomes. For resequencing of mutants and correction of possible PacBio sequencing errors in the reference genomes, DNA samples were prepared according to Nextera© XT DNA Library Preparation Guide (Illumina Inc.), and sequencing was performed using a MiSeq™ Desktop Sequencer (Illumina Inc.). The raw data were analyzed in CLC Genomic Workbench v.20 (Qiagen), where raw reads were trimmed and aligned to the reference sequences to analyze the genetic changes. Genetic changes were detected using the Microbial genomics module and resequencing SNP and InDel identification software in the CLC Genomics workbench. For clones, the cut-off for calling SNPs was set to >75% (75% of the reads have the SNP), and for InDels, a frequency of >50% among reads was used to compensate for the inherent difficulty of correctly aligning reads with InDels to the reference sequence. Each genetic change was manually inspected to verify the correctness. Genetic changes with at least 20% frequency were included for populations unless lower-frequency mutations known to be present in clones were found. Protein sequences were further analyzed with BLASTp (NCBI) and a conserved domain database[85] for functional determination. Clone and population strain numbers from the evolved biofilm lineages with information on all mutations are noted in Supplementary Data 2. Sequence, capsule locus, and O antigen locus types were determined using the Kaptive database[86].

## Transmission electron microscopy

Bacterial strains were grown at 37 °C with shaking (180 rpm) overnight in BHI and 2 ml centrifuged at 8000xg for 5 min. The supernatant was removed, 5 ml of ice-cold fixative solution (2.5% glutaraldehyde and 1% paraformaldehyde in 0.1 M sodium cacodylate buffer) was added, and the fixation continued overnight at 4 °C. Further sample preparation and imaging were done at the BioVis EM node (Rudbeck's laboratory, Uppsala University). The samples were post-fixed with 1% osmium tetraoxide in 0.1 M PIPES buffer and dehydrated in increasing ethanol concentrations (70%, 90%, and 100%). After dehydration, the samples were either negatively stained with uranyl acetate or further embedded in Agar 100 resin, ultrathin-sectioned (60 nm thickness), contrasted with uranyl acetate and lead citrate, and air-dried. The samples were imaged using a transmission electron microscope FEI Technai G2 at 80 kV.

## Scanning electron microscopy

The biofilms grown on silicone-coated pegs were fixed at 16 h or 48 h and prepared for SEM imaging as described before[23]. Images in Fig. 6e were color-modified in Adobe Photoshop to distinguish the biofilm-covered area from the peg surface. Raw images are available in the Source data file.

## Sedimentation resistance assay

Bacteria were grown in BHI to late stationary phase, and cultures were centrifuged for 5 min at 1000 × $g$. The sedimentation constant was calculated as the ratio of $OD_{600}$ of the supernatant and the $OD_{600}$ before centrifugation[44].

## Uronic acid assay

Non-attached and total extracellular polysaccharides were determined by the uronic acid assay as described in ref. [87]. In short, bacteria were grown at 37 °C with agitation (190 rpm) in low-salt LB medium overnight, and the optical density at 600 nm was recorded for normalization. Extracellular polysaccharide was purified by adding 250 μl of overnight culture to 50 μl of 1% Zwittergent 3-14 in 100 mM citric acid, pH 2, followed by incubation at 50 °C for 20 min. In parallel, non-attached polysaccharides were purified by the addition of 50 μl of water instead of detergent. Samples were centrifuged at 17,000 × g for 5 min at room temperature, and 100 μl polysaccharide-containing supernatant was transferred to 400 μl absolute ethanol and incubated on ice for 20 min for precipitation. Polysaccharides were pelleted by centrifugation at 17,000 × g for 5 min at 4 °C, pellets were dried at room temperature for 10 min, and re-suspended in 200 μl water and incubated at 37 °C for 30 min to fully solubilize.

Quantification of polysaccharides was done by the addition of 1.2 ml of 0.0125 M sodium tetraborate in concentrated sulfuric acid to each 200 μl of purified polysaccharide. Samples were incubated at 100 °C in a heat block for 5 min, and then cooled on ice. 10 μl of 0.3% (3 mg/ml) w/v 3-phenylphenol in 0.5% (0.125 M) NaOH was added, and the absorbance at 520 nm was measured. Relative polysaccharide content was calculated by subtraction of the difference in A520 before and after the addition of 3-phenylphenol from blanks, normalization based on A600 of the original overnight cultures, and expressed as relative values for the mutants compared to the parental strain.

## mrkA and bcsA mRNA quantification by qPCR

Bacteria were grown with agitation (190 rpm) in BHI, and culture aliquots for RNA extraction were taken during mid-exponential (OD = 0.5), late exponential (OD = 1.0), and early stationary phase (6 h of growth). RNA was extracted using the RNeasy Mini Kit (Qiagen), and genomic DNA was removed using the TURBO DNA-free kit (Invitrogen). Complementary DNA (cDNA) was synthesized using the High-Capacity Reverse Transcriptase kit (Applied Biosystems) with approximately 500 ng of extracted RNA. The expression of mrkA and bcsA was assessed with qPCR (Illumina Eco Real-Time PCR System) using the SYBR Green Master mix (Thermo Fisher) relative to the expression of the glnA reference gene[88]. Three technical replicates were used for each biological replicate in qPCR to determine $C_T$ values. Primer sequences can be found in Supplementary Data 3. The mRNA levels were normalized to the respective parental strains' mRNA levels for the respective conditions using the $2^{-\Delta\Delta C_T}$ method[89].

## c-di-GMP quantification by LC-MS

Selected clones were grown in BHI with shaking (180 rpm) at 37 °C until the mid-exponential phase (OD = 0.5) or early stationary phase (6 h of growth), and culture aliquots (concentrated to OD 5) were washed three times in ice-cold sterile 1x PBS. The samples were kept at −80 °C until the extraction was done at the Swedish Metabolomics Centre (Umeå, Sweden). Frozen samples were extracted using 400 μl 80% MeOH + salicylic acid-D6 (2 pg/μl). Supernatant (5 μl) from the centrifuged samples was injected into the instrument (Agilent 6490 triple quadrupole system). To make a calibration curve, c-di-GMP was serially diluted to produce a range from 31 fg/μl to 8 pg/μl. The amount of c-di-GMP was expressed as the mass of c-di-GMP (pg) per $10^9$ CFU in a sample.

## Survival in human serum

Selected clones were grown overnight in LB, diluted 10.000-fold in PBS, and approximately $10^5$ CFU mixed with human serum from male AB plasma (Sigma-Aldrich, product number H4522) in a total volume of 200 μl. Due to natural batch variation in human serum activity, the assays were performed either with undiluted serum or 40% dilution (in PBS) to achieve the same killing effect relative to the parental strains. The serum was heat-inactivated for 30 min at 56 °C for negative control experiments. The CFU counts were performed at 0 h and after 3 h of static incubation at 37 °C. For testing the biofilm's sensitivity to human serum, biofilms were formed on FlexiPeg for 48 h, washed in PBS, and exposed to human serum for 3 h, after which CFU/peg was determined.

## Virulence in *Galleria mellonella* larvae

Selected mutants were tested for the killing of *Galleria mellonella* larvae. Larvae were purchased from Herpers choise (Uppsala, Sweden) and stored for a maximum of 3 days at room temperature before experiments. Only healthy larvae (no discoloration, active) of similar size (approximately 250-300 mg) were used for the experiments. Bacterial cultures grown overnight in BHI were diluted in PBS and 10 μl injected via the top right proleg using a 25 μl Hamilton 7000 syringe (Model 702 RN, 22S gauge needle). For each strain, 20 larvae (1 biological replicate per 10 larvae) were injected and monitored hourly for 7-12 h post-injection on the first day, then at 24 h, 36 h, and 48 h post injection during incubation at 37 °C. Aliquots from dilutions were plated for CFU counts to confirm the correct infectious load. Along with experimental larvae, 10 larvae were injected with 10 μl of sterile PBS to account for possible injection trauma, and 10 larvae served as non-injection control. The larvae were considered dead when not moving after stimulation with a pipette tip. In addition, larvae were given health index scores at 8 h, 12 h, 24 h, and 48 h post injection based on the previously described scoring system, considering melanization status and changes in movement/responsiveness[90]. A cumulative score was calculated for each replicate by adding the scores for each individual larva at different timepoints. Survival curves were analyzed using the log-rank (Mantel-Cox) test in GraphPad Prism version 9.2.0 for macOS (GraphPad Software, San Diego, CA, USA).

## Mammalian cell culture

T24 bladder transitional carcinoma cells (ATCC HTB-4) were grown in McCoy's 5a Medium Modified (Sigma) supplemented with 10% heat-inactivated fetal bovine serum (FBS). A549 lung carcinoma cells (ATCC CCL-185) were grown in RPMI 1640 (Gibco) containing 10% heat-inactivated FBS. Epithelial cells were seeded in tissue culture-treated 24-well plates 18-48 h prior to infection.

## Adherence to human epithelial cells

T24 and A549 epithelial cells were seeded densely enough to cover the entire bottom of the well and thereby limit bacterial binding to the plastic. Bacteria were grown in 2 ml of BHI broth overnight with shaking (180 rpm) at 37 °C. The following day, cultures were used directly for infection with stationary phase bacteria or diluted 1:100 in BHI broth and grown with shaking at 37 °C for 1.5 h for infection with exponentially growing bacteria. Each well was infected with $10^7$ bacteria. To synchronize infection, the bacteria were centrifuged onto the cells at 250xg for 5 min and left to adhere for 30 min at 37 °C and 5% $CO_2$. Cells were washed three times in PBS to remove unbound bacteria. The monolayers were solubilized in 0.2% sodium deoxycholate (DOC), and serial dilutions were plated on LA plates to enumerate the adhering bacteria.

## Statistical analysis

The respective analysis is specified for each experiment and figure in the figure legends. Biological replicates in all experiments denote independent cultures grown from single colonies.

## Reporting summary

Further information on research design is available in the Nature Portfolio Reporting Summary linked to this article.

## Data availability

The complete genome sequences of IA565 (DA11912) and C3091 (DA12090) have been deposited under BioProject PRJNA473315 and PRJNA473316, respectively, and the outbreak index isolate under PRJNA857654. Sequence files for all mutants in this study have been deposited in the Short Read Archive (NCBI) under BioProject PRJNA1048869. Source data are provided with this paper. Protein structure data used were: Wzc (7NHR) and MrkD (https://alphafold.com/entry/AF-P21648-F1). Videos are provided via figshare (https://doi.org/10.6084/m9.figshare.29816147) Source data are provided with this paper.

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

## Acknowledgements

Parts of this work were presented at ASM Microbe 2019 (San Francisco, USA), EuroBiofilms 2019 (Glasgow, UK), and the 8th National Infection Biology/Microbiology Meeting 2019 (Bålsta, Sweden). The authors would like to thank Victoria Sternhagen for performing the SEM analysis and data acquisition carried out within the Uppsala University academic cleanroom, a member of the MyFAb national research infrastructure. Monika Hodik from BioVis (Rudbeck Laboratory, Uppsala University) is acknowledged for her help with TEM sample preparation and imaging. Alice Andersson is acknowledged for experimental support. The Swedish Metabolomics Centre (Umeå, Sweden) is acknowledged for the quantification of c-di-GMP by LC-QqQ-MS. Uppsala Antibiotic Center is acknowledged for funding P.C. as a PhD student within the UAC research school. We also thank Dan I. Andersson and Vaughn S. Cooper for their comments on the original manuscript draft. Uppsala Antibiotic Center is acknowledged for funding P.C. as a PhD student within the UAC research school, and the Carl Tryggers Foundation for research funding to LS.

## Author contributions

G.Z. and L.S. conceived the study and designed experiments. G.Z. performed experimental evolution, screening of clones, extracted genomic DNA from clones and populations, competitions with evolved clones, prepared samples for SEM and TEM imaging, c-di-GMP quantification, performed capsule extractions, sedimentation assays, *G. mellonella* infection assays, and strain constructions. G.Z. and P.C. measured biofilm capacity, early attachment to pegs, extracted RNA, and performed qPCR, serum killing assays, cellulase assays, and exponential growth rate measurements. M.W. performed experiments with epithelial cells and uronic acid assay. P.C. and M.W. performed mutation rate experiments. G.Z. and L.S. performed bioinformatic analysis. G.Z. and L.S. analyzed data and wrote the manuscript. G.Z. made the figures. All authors read, edited, and approved the final version of the manuscript.

## Funding

## Competing interests

The authors declare no competing interests.
