## [Transparent Peer Review file · Nature Communications]

Rapid evolution of *Klebsiella pneumoniae* biofilms *in vitro* delineates adaptive changes selected during infection

Corresponding Author: Professor Linus Sandegren

Version 0:

Reviewer comments:

Reviewer #1

(Remarks to the Author)

The noteworthy results in this manuscript are predominantly related to the biofilm ecological niche and the propensity for mutation in those bacteria present in biofilms. The ?increased probability of mutation is not explained, though cell density is one candidate. We are 'left in the dark' on the relative mutation rates of planktonic bacteria – are these equally genetically 'plastic' as the counterpart bacteria present in biofilms. This important would reveal whether the higher mutation rates seen in key biofilm components/proteins e.g. the emergence of hypermucooid and other mutants (mrkJ, mrkD etc) are caused by selective pressure under biofilm growth per se and thus likely to comes from factors like 'overcrowding' and competition for nutrients.

The authors reference a Nature Comms paper in the discussion (#74) that also saw hypermucooid mutants appear in an evolution experiment, but which used serial passaging of planktonic cells – and these mutants were also better biofilm formers. Hence, in Ref #74m forming a biofilm wasn't a driver for mutagenesis.

The manuscript currently reads more like a thesis than a tight paper describing a very interesting phenomena. There are (for me) too many messages in the manuscript – the story would be stronger if some of the minor results were removed (perhaps to Supp figures or entirely) to add some better 'linearity' to the storyline, and the manuscript focussed on the key issue of bacterial evolution during biofilm formation. The addition of so many points can leave the reader confused about what constitute the main message(s).

The paper raises issues about what is really the 'wild type' sequence in an organism where sampling bias in the sequencing databases is very strong. We often take a very anthropomorphic view of *Klebsiella* – the sequence databases are consequently dominated by human disease cases, and environmental isolates are grossly under-represented. If the 'mutation' of key genes in biofilm formation happens so readily, are the majority of sequences in the databases reflective of biofilm adapted- or non-adapted genotypes? Fundamentally, what is a mutant and what is wild type?

The work will likely be well cited. Biofilms are a common focus for microbiologists and materials scientists and a new understanding that biofilm reduction approaches may also help limit mutation rates in an era of rising AMR will be enthusiastically supported by those advocating different approaches to biofilm inhibition.

There is a lot known about regulation of the key adhesin the Mrk fimbriae, and the genomes of *Klebsiella* reveal typically >10 enzymes capable of producing c-di-GMP (i.e. guanylate cyclases) and a similar number of enzymes (PDEs) that reduce concentrations of the key co-factor in Mrk regulation. YfiN is likely not the only contributor to [c-di-GMP].

I have included a number of questions that address methodological issues.

Questions

1. The biofilm pegs were vortexed to enable counting – were there still bacteria attached, i.e. was the analytical bias towards more loosely adherent organisms?
2. How many separate i.e. 'biological' repeats on different days were there in the sequencing analysis (Fig 2) – for example, were the same genes identified each time using different starter cultures? This is a key issue since the paper really rests on

this mutant frequency data.

3. In the Galleria model, what does infection sensitivity represent in so-called WT isolates? Is there a correlation between killing of the wax moth and systemic infection in humans, for example? Is the model validated as a proxy for human disease or like the murine model, demonstrate virulence linked with capsule type?
4. Is the biofilm that occurs in hypermucooid strains the same as that in normally encapsulated Kp? – there is evidence that the de-encapsulated mutants adhere more efficiently to biotic and abiotic surfaces than normally encapsulated strains.
5. YfiN is one of a number of diguanylate cyclases in Kp (bioinformatics suggests 14 or more in some isolates, married with 9 or more PDEs) – was there evidence of DGC compensations in YfiN mutants – e.g. mutations in other GGDEF-containing proteins i.e. to maintain [c-di-GMP] sufficient to enable Mrk expression at levels that will enable biofilm formation?
6. The decrease of Mrk expression in static cultures – was the analysis performed on planktonic cells or both adherent and planktonic cells?
7. The colony phenotype needs more explanation for a general audience and data/citations suggesting how the colony morphotype is linked with any of the mutations observed would be helpful. Is it due to increased cellulose production as hinted? A similar explanation was given for the Rho mutant in the results (from line 275), but not for the GGDEF domain protein mutant (DL426_09630).
8. For the mrkD mutants, is it known that mrkA is expressed at similar levels in these isolates?
9. The Figure 4. c and d legends and the results text need to be swapped around to match the images.

Reviewer #2

(Remarks to the Author)

Reviewer #3

(Remarks to the Author)

The manuscript uses experimental evolution of three *K. pneumoniae* isolates to identify adaptive changes relevant to biofilm growth in this pathogen. They identify some convergent genetic changes, accompanied by varied phenotypic changes. Some of the genes and pathways that changed during experimental growth overlap with mutations attributed in a separate study to adaptive evolution in vivo during extra-intestinal infections in humans.

General comments:

1. The work presents a large amount of lab experiments, but the statistical analysis does not do them justice. In addition, the data presentation prevents reader from linking different parts of analysis. For example, in Fig 4d, the reader has to mentally align rows for each parental strain to see if same mutation in different strains leads to the same outcome. Same applies to Fig 4b and 4e and other figures. Statistical tests are used inconsistently. For example, Fig 6e does not have any, but Fig 6f does.
2. A lot of conclusions are not backed by any statistical tests which limits their robustness. As a reader, I find it hard to know the sample sizes, distributions, etc. Given the large amount of experimental results, some adjustment for multiplicity of testing is also necessary.

Specific comments:

1. Figure 2a: In contrast to the rest of the figures in Figure 2, 2a doesn't split samples by progenitor isolate (IA565, C3091, DA14734). Without this, it's unclear if the change is evenly distributed among clones or due to single clone. Lines 98-101: I think this is based on mutation count in Figure B? These mutations are in the core genome (based on Methods), the accessory genome is not taken into account? Similarly "clones had a more diverse mutation spectrum" also requires statistical support as it's not evident from Fig. 2d.
2. Lines 106: How is the phenotype "hypermucoid" defined?
3. Line 107: In the supplementary data (Table S1), the wzc mutations are odd. For DA14734_full, silicon has 2 lineages with wzc mutations, silicon + fibrinogen has 5, but only two different alleles. HT (low inoculum) has 7 lineages with wzc mutations, but only 5 of them are the same finally HT (high inoculum) has 4 lineages with mutations and three alleles. Using HT (low inoculum) as example, are the authors assuming the mutation arose twice at cycle 4 or seven times at cycle 5? Also, cycle 2 is listed in silicone+fibrinogen, but all other media only show cycle 5 or 6. Does that mean there were no mutations for cycles 1-4?
4. Lines 116-117: I'm not sure how these sweeps can be inferred from the cited figures. For Figure 1D, replacing "Morphotype frequency %" with count of samples with morphotypes would be more helpful.
5. Table S1: this table is extremely difficult to understand and appears to have a lot of internal notes. If possible, this table should be cleaned-up as it's very important for understanding the manuscript.
6. Line 136: What is the data for this test?
7. Line 175: Was rcsDG217R was the only pangenome difference between that isolate and parental strain?

8. Fig 4b: missing p-values., annotation for (c) and (d) appears the wrong way round.

9. Fig 6e: is this reduction relative to same isolate without cellulase or relative to parental isolate? The implications of this figure on Line 261-264, while accurate, give an impression that these mutations have functional implications, but data doesn't demonstrate this given strain effect and lack of shared mutations in presented strains.

10. Line 265: The conclusion about mode of mediation can be made if supported either by targeted experiments or statistical model that relates the data in (b), (c), (e) and (f) of Fig. 6.

11. Line 279-280: The data does suggest that cellulase effects biofilm formation, but the inconsistency of results in Fig 6e demonstrates that cellulase does not always inhibits biofilm formation.

12. Lines 440-441: How many mutations were accumulated in each cycle in each lineage for different parental strains? This can be demonstrated via a box and whiskers plot with a series for each parental strain and whiskers showing range of mutations across lineages at each cycle. Evidence for Line 497 could also be added to this plot.

13. Lines 507-510: I cannot decipher what this means "For clones, the cut-off for calling SNPs was set to >75%, and for InDels, a frequency of >0,5 among reads was used, and each change was manually inspected." or why different cut-offs were used; or why the thresholds are presented by a mix of % and proportion in the same sentence (75% vs 0.5). Please clarify.

Reviewer #4

(Remarks to the Author)

The group of Sandegren conducted a study investigating the evolution of biofilms in three strains of *Klebsiella pneumoniae*, "simulating" catheter-associated infections through surface-attached biofilms. They observed rapid evolutionary changes affecting capsule formation, fimbrial adhesion, and c-di-GMP-dependent pathways, with results sometimes varying among strains. The study also found that acute virulence was associated with specific genetic changes rather than overall biofilm capacity. Additionally, the authors noted overlaps between the changes and previously identified adaptations in urinary tract and wound isolates. Although the study's goals are clearly outlined and its presentation is satisfactory, there are certain issues that need to be addressed before it can be published in Nature Communications.

Major points

- My primary concern is the limited variety of strains and strain diversity in the study. It would significantly enhance the study's relevance to include a broader range of strains, such as hypervirulent or convergent representatives, to yield more meaningful biological outcomes, especially in the context of capsular adaptations. While the authors did find mutational overlaps with clinical strains and changes documented in the literature, expanding the strain selection would strengthen the study's overall impact and applicability. With that being said, if I understand correctly, the overlap with previous isolates pertain to strains from a clonal outbreak, which typically exhibit comparable genetic compositions. Therefore, it's possible that the number of identical mutations is biased.
- The novelty of both the findings and the applied methodologies seems somewhat limited. Many of the observed changes, as acknowledged by the authors, have been previously documented and are expected. Additionally, the FlexiPeg device has been previously published. As mentioned before, to increase interest, it would be beneficial to include other genetic backgrounds of *K. pneumoniae*, enabling a more comprehensive exploration of commonly applicable patterns. The study could have also been elevated to the next level by investigating the combination of mutations leading to various phenotypes and exploring the underlying mechanisms of compensatory processes.

Medium points

- The degree to which the device mimics the actual clinical situation should be elucidated further. How closely does the device replicate the clinical scenario?
- L. 103-104: potential trade-offs should be addressed further, e.g. in additional fitness/virulence assays.

Minor points

- L. 34-35: this is not true for hypervirulent *K. pneumoniae*
- L. 328: hypervirulent or classic *K. pneumoniae*?
- L. 433: which medium has been used?
- Why sometimes 10 and sometimes 6 individual lineages in the experimental evolution?
- I believe that the sedimentation assay is solely referenced in the Materials and Methods section.
- Clarify the rationale behind the relatively short duration of the experimental set-up (48 hours max.).

Version 1:

Reviewer comments:

Reviewer #1

(Remarks to the Author)

I have limited my comments to Reviewer 1 though agree with many of the points raised by the other reviewers, particularly the use of statistics and biological rather than technical replicates to demonstrate the phenotypes identified. I have asked my assistant reviewer to also review the changes.

Point 1 – mutation rates in biofilm and planktonic bacteria

Reviewer: The noteworthy results in this manuscript are predominantly related to the biofilm ecological niche and the propensity for mutation in those bacteria present in biofilms. The increased probability of mutation is not explained, though cell density is one candidate. We are 'left in the dark' on the relative mutation rates of planktonic bacteria – are these equally genetically 'plastic' as the counterpart bacteria present in biofilms. This important would reveal whether the higher mutation rates seen in key biofilm components/proteins e.g. the emergence of hypermucoid and other mutants (mrkJ, mrkD etc) are caused by selective pressure under biofilm growth per se and thus likely to come from factors like 'overcrowding' and competition for nutrients.

Authors: We appreciate that the ecological dynamics within biofilms, including factors such as cell density and nutrient competition, are likely to play a significant role in shaping the emergence of adaptive mutants. The question of whether mutation rates differ between planktonic and biofilm-associated bacteria is indeed important, though it falls outside the scope of our current study.

Our experimental evolution approach was specifically designed to enrich for biofilm-forming variants by repeatedly removing planktonic and loosely attached cells. As such, the mutants we observe—such as those affecting mrkJ and mrkD—are likely the result of selection for traits that enhance biofilm formation, rather than an inherently increased mutation rate within the biofilm environment. These mutations may arise in planktonic cells that subsequently attach and persist, or during biofilm development itself.

Reviewer: The manuscript describes mutations present in bacteria in biofilms. The Reviewer has asked the simple question whether the mutation rate is the same or different from planktonic bacteria. There are relatively straightforward measures of mutation rates involving reconstitution of resistance or metabolic deficiencies in selectable single nucleotide point mutants on defined media, and it is not clear why the authors cannot apply these techniques to measuring the rates in biofilms and when the bacteria are in solution. The constant removal of planktonic bacteria is likely to select for rare bacteria that have lost the ability to adhere through e.g. the Type 3 Mrk fimbriae. I feel that this question should be addressed with experimental data.

Point 2 – structured environments and mutation rates

Reviewer: The authors reference a Nature Comms paper in the discussion (#74) that also saw hypermucoid mutants appear in an evolution experiment, but which used serial passaging of planktonic cells – and these mutants were also better biofilm formers. Hence, in Ref #74m forming a biofilm wasn't a driver for mutagenesis.

Authors: In the study in Ref. 79 (previously Ref. 74), the authors did not perform serial passaging of planktonic cells; instead, cultures were statically grown in 24-well plates to produce "structured environments," as stated by the authors themselves. Therefore, the argument for selection in biofilms stands as described above, but it is not an increase in mutation rate per se, but rather a function of selection within the biofilm niche.

Reviewer: Evolutionary experiments are very tough to control and while the authors have provided a good explanation to counter the Reviewers comments, if the mutation rate is indeed elevated in biofilms, this is something that should be tested and reported.

From line 429: "Our results suggest that biofilm growth serves as a selective pressure for the emergence of both the hypermucoid and capsule-loss mutants, which we also identified as within-host-selected infection site adaptations during the Uppsala hospital outbreak. In parallel with our work, another study reported the selection of wzc mutants in structured environments, which was achieved by serial passaging in statically incubated 24-well plates."

In the Nat comms paper (actually Ref. 77), they used the term 'structure' to refer to limited dispersal and localised gradients in statically grown cultures. But the cells passaged from this part of the culture is still technically planktonic (free floating) and certainly not from the biofilm. So there was no intentional surface selection or maintenance of attached biomass. This point could be clarified (from line 429 in the manuscript), because the way they've tied in their results it reads that 'structured environments' equates to biofilm for someone who hasn't read that cited paper, and it's not – it's a fancy term for planktonic. So from their response I'm unsure how it all argues for selection in biofilms vs increases in mutation rates, because that hasn't been tested.

Point 3 – reads like a thesis

The authors have addressed this issue and tightened the writing of the paper

Point 4 – wild type versus parental

The term parental is appropriate – from sequencing of isolates alone it is impossible to establish what is 'wild type' and what

is a mutant, where there is opacity about the individual lineages, and the sequences of the isolate(s) from which they were derived.

Point 5 – multiple PDEs and DGCs

the authors acknowledge that there is considerable complexity in the bacterial 'management' of [c-di-GMP] levels, especially in the signals that activate and regulate the multiple guanylate cyclases and PDEs. The ligands and environmental conditions necessary for the activation of these enzymes remains cryptic for the most part but clearly paints a picture of complexity when [c-di-GMP] is a key component of Mrk expression.

Question 1

This is satisfactorily answered.

Question 2

This is satisfactorily answered. The addition of further experimental detail provided in the resubmitted manuscript is welcomed.

Question 3

This is satisfactorily answered and the text updated.

Question 4

Is there some sort of quantitative data that might accompany "yield distinct phenotypes across strains" so that others who repeat these experiments can identify the phenomena?

Question 5

This is a fascinating observation but it is not clear whether there was a parental strain from early in the outbreak that had different sequences for the ? 4 changed GGDEFs identified in the outbreak strain. Is it evidence of previous mutation and selection or did it occur during the outbreak?

Question 6 – Mrk expression in static cultures

The authors make a clear and honest point about the selection that can go on for hyper fimbriated isolates that aggregate and therefore are difficult to analyse. If this is clearly stated in the revised manuscript then there this is probably all that can be done.

Question 7 – colony phenotypes

This is satisfactorily answered and the text updated.

Question 8

This is satisfactorily answered and the text updated.

Question 9

This is satisfactorily answered and the text updated.

Reviewer #3

Use of statistics – this should be consistent and the same tests used, where possible.

Measuring hypermuroid – there are simple tests like uronic acid levels in solution, which can quantitatively determine whether the bacteria are hypermuroid.

Reviewer #2

(Remarks to the Author)

REVIEWER COMMENTS

Reviewer #1 (Remarks to the Author):

The noteworthy results in this manuscript are predominantly related to the biofilm ecological niche and the propensity for mutation in those bacteria present in biofilms. The increased probability of mutation is not explained, though cell density is one candidate. We are 'left in the dark' on the relative mutation rates of planktonic bacteria – are these equally genetically 'plastic' as the counterpart bacteria present in biofilms. This important would reveal whether the higher mutation rates seen in key biofilm components/proteins e.g. the emergence of hypermuroid and other mutants (mrkJ, mrkD etc) are caused by selective pressure under biofilm growth *per se* and thus likely to come from factors like 'overcrowding' and competition for nutrients.

We appreciate that the ecological dynamics within biofilms, including factors such as cell density and nutrient competition, are likely to play a significant role in shaping the emergence of adaptive mutants. The question of whether mutation rates differ between planktonic and biofilm-associated bacteria is indeed important, though it falls outside the scope of our current study.

Our experimental evolution approach was specifically designed to enrich for biofilm-forming variants by repeatedly removing planktonic and loosely attached cells. As such, the mutants we observe—such as those affecting mrkJ and mrkD—are likely the result of selection for traits that enhance biofilm formation, rather than an inherently increased mutation rate within the biofilm environment. These mutations may arise in planktonic cells that subsequently attach and persist, or during biofilm development itself.

The authors reference a Nature Comms paper in the discussion (#74) that also saw hypermuroid mutants appear in an evolution experiment, but which used serial passaging of planktonic cells – and these mutants were also better biofilm formers. Hence, in Ref #74m forming a biofilm wasn't a driver for mutagenesis.

In the study in Ref. 79 (previously Ref. 74), the authors did not perform serial passaging of planktonic cells; instead, cultures were statically grown in 24-well plates to produce "structured environments," as stated by the authors themselves. Therefore, the argument for selection in biofilms stands as described above, but it is not an increase in mutation rate *per se*, but rather a function of selection within the biofilm niche.

The manuscript currently reads more like a thesis than a tight paper describing a very interesting phenomena. There are (for me) too many messages in the manuscript – the story would be stronger if some of the minor results were removed (perhaps to Supp figures or entirely) to add some better 'linearity' to the storyline, and the manuscript focussed on the key issue of bacterial evolution during biofilm formation. The addition of so many points can leave the reader confused about what constitute the main message(s).

We have revised the manuscript's structure to make the storyline more straightforward for the reader to follow. The sections have been divided to focus on the key aspects of the

evolutionary outcome of the selections, evolutionary trajectories of phenotypes and genotypes, and the characterization of the three most dominant mutational targets (capsule, fimbriae, c-di-GMP regulation). We hope this makes the story more readable. Nonetheless, the study is based on a large number of different phenotypic tests to decipher what constitutes selective factors in the biofilm and how the genetic changes selected affect the pathoadaptability of the bacteria.

The paper raises issues about what is really the 'wild type' sequence in an organism where sampling bias in the sequencing databases is very strong. We often take a very anthropomorphic view of *Klebsiella* – the sequence databases are consequently dominated by human disease cases, and environmental isolates are grossly under-represented. If the 'mutation' of key genes in biofilm formation happens so readily, are the majority of sequences in the databases reflective of biofilm adapted- or non-adapted genotypes? Fundamentally, what is a mutant and what is wild type?

We share the reviewer's opinion that "wild-type" is a constructed concept that cannot be reliably applied to clinical isolates, and we therefore do not use it. Instead, we use the designation "parental" for the three strains that we start the evolution with. This merely reflects their ancestral state in this particular evolution experiment. *Klebsiella* sequences in the databases are indeed very dominated by clinical isolates, although this trend is now changing as more environmental isolates are being added. The three strains we chose to start with are well-characterized and were therefore selected as starting points for our study. The rapid selection of better biofilm formers indeed raises the question of how often this occurs during patient colonization/infection and, therefore ends up in the sequencing databases. This is one of the essential aspects that we address in our study by describing the genotype-phenotype connections in selected mutants and comparing them to clinical isolates with/without increased biofilm capacity. The use of the outbreak clone enables us to make very detailed comparisons to selection during colonization and infection of patients.

The work will likely be well cited. Biofilms are a common focus for microbiologists and materials scientists and a new understanding that biofilm reduction approaches may also help limit mutation rates in an era of rising AMR will be enthusiastically supported by those advocating different approaches to biofilm inhibition.

We also hope that this work will shed further light on the evolution of biofilm formation and its role in *K. pneumoniae* infections.

There is a lot known about regulation of the key adhesin the Mrk fimbriae, and the genomes of *Klebsiella* reveal typically >10 enzymes capable of producing c-di-GMP (i.e. guanylate cyclases) and a similar number of enzymes (PDEs) that reduce concentrations of the key co-factor in Mrk regulation. YfiN is likely not the only contributor to [c-di-GMP].

We agree with the reviewer and are aware of the multiple phosphodiesterases and diguanylate cyclases present on *Klebsiella* genomes, including the strains studied here. We do not propose that YfiN is the only contributor to fimbrial regulation, but we understand that the original text in this section might have been confusing in depicting YfiN as the only

DGC. We have modified the text and the Figure (now Fig. 8) accordingly to avoid confusion and discuss the additional c-di-GMP-associated proteins further. We have also included targeted deletions to assess their phenotypic effects.

I have included a number of questions that address methodological issues.

Questions

1. The biofilm pegs were vortexed to enable counting – were there still bacteria attached, i.e. was the analytical bias towards more loosely adherent organisms?

We have previously confirmed that high-speed vortexing is an efficient method for removing biofilms from the pegs, and we compared it to other forms of removal (Zaborskyte et al., Modular 3D-Printed Peg Biofilm Device for Flexible Setup of Surface-Related Biofilm Studies. *Frontiers in cellular and infection microbiology*, 1407 (2022)). Anything that is “loosely adherent” is washed away before disrupting the population attached to the peg, and no or very little biological material remains on the pegs after vortexing.

2. How many separate i.e. 'biological' repeats on different days were there in the sequencing analysis (Fig 2) – for example, were the same genes identified each time using different starter cultures? This is a key issue since the paper really rests on this mutant frequency data.

We used multiple (6-10) independent lineages (starter cultures from individual bacterial colonies) for each strain and each condition to address genetic diversity in evolving populations. This way, the evolution in each lineage will be separate from the others, and repeated isolation of the same mutations or mutational combinations indicates the evolutionary success of them. Here, the bottlenecks during passaging play a crucial role in allowing even rare mutations to be passed on. High-frequency mutations will usually dominate an evolving population early if they provide a sufficient fitness advantage in the selective landscape; however, a less frequent mutant with higher fitness may outcompete them in the long run, as observed for several lineages. This is now illustrated in the new Figure 5. The number of lineages for each strain and each condition is listed in Supplementary Data 1. Morphotype frequencies were calculated based on at least 100 colonies. Sequencing was done on single clones from multiple lineages.

3. In the Galleria model, what does infection sensitivity represent in so-called WT isolates? Is there a correlation between killing of the wax moth and systemic infection in humans, for example? Is the model validated as a proxy for human disease or like the murine model, demonstrate virulence linked with capsule type?

By varying the bacterial inoculum, the *G. mellonella* model system can accommodate both very low virulence isolates as well as high virulence. Throughout this study, we have used comparisons with the same inoculum for parental strains and their derived mutants to verify differences in virulence. For example, IA565, which is basically avirulent in mice and humans (Lau et al., 2007, *Microb. Pathog.* 42, 148–155), is also attenuated in Galleria. *G. mellonella* has been extensively validated and has an innate immune system with both cellular and

humoral responses similar to those of vertebrates, and is the preferred infection model for cKp, which do not perform well in mice models in contrast to hypervirulent Kp (Browne, et al., *An analysis of the structural and functional similarities of insect hemocytes and mammalian phagocytes. Virulence* **4**, 597–603 (2013). Tsai, et al., *Galleria mellonella infection models for the study of bacterial diseases and for antimicrobial drug testing. Virulence* **7**, 214–229 (2016). Wand, et al., *Complex interactions of *Klebsiella pneumoniae* with the host immune system in a *Galleria mellonella* infection model. Journal of Medical Microbiology* **62**, 1790–1798 (2013). Insua, J. L. et al. *Modeling *Klebsiella pneumoniae* Pathogenesis by Infection of the Wax Moth *Galleria mellonella*. Infection and Immunity* **81**, 3552–3565 (2013). Bruchmann, et al., *Identifying virulence determinants of multidrug-resistant *Klebsiella pneumoniae* in *Galleria mellonella*. Pathogens and disease* **79**, ftab009 (2021)).

We have updated the description of the infection experiments to clarify this.

4. Is the biofilm that occurs in hypermucoid strains the same as that in normally encapsulated Kp? – there is evidence that the de-encapsulated mutants adhere more efficiently to biotic and abiotic surfaces than normally encapsulated strains.

No, they are quite different; this is what we describe in the section “Various levels of mucoviscosity enhance biofilm formation but yield distinct phenotypes across strains”. We have updated the section on how mucoviscosity affects the biofilm formation to clarify this.

5. YfiN is one of a number of diguanylate cyclases in Kp (bioinformatics suggests 14 or more in some isolates, married with 9 or more PDEs) – was there evidence of DGC compensations in YfiN mutants – e.g. mutations in other GGDEF-containing proteins i.e. to maintain [c-di-GMP] sufficient to enable Mrk expression at levels that will enable biofilm formation?

This is an interesting question; however, we did not find any mutants with combined mutations in more than one PDE/DGC gene in the experimental evolutions. However, during the hospital outbreak (Ref. 22), we identified isolates with mutations in up to four different EAL/GGDEF genes. It is possible that if we had prolonged the *in vitro* evolution experiment such combinations could have occurred. Another thing to note is that none of the GGDEF domain protein genes, including *yfiN*, had mutations in the GGDEF domain itself, therefore, we would not expect them to abolish DGC function but rather alter it. We also constructed a deletion mutant of the uncharacterized GGDEF gene DL426_RS09305, which did not confer an increase in biofilm or the wrinkly morphotype associated with the mutation A203V selected during evolution.

6. The decrease of Mrk expression in static cultures – was the analysis performed on planktonic cells or both adherent and planktonic cells?

Analysis was conducted in shaking cultures (exponential and early stationary), but mutants with a high increase in biofilm start aggregating very early on, so they are not entirely planktonic even under these conditions. This has been clarified in the Materials and Methods section.

7. The colony phenotype needs more explanation for a general audience and data/citations suggesting how the colony morphotype is linked with any of the mutations observed would be helpful. Is it due to increased cellulose production as hinted? A similar explanation was given for the Rho mutant in the results (from line 275), but not for the GGDEF domain protein mutant (DL426_09630).

We have now extended the descriptions of the morphotypes as a separate Results section to explain the genetic changes and biofilm phenotypes. They are also part of detailed follow-up experiments in later parts of the manuscript, and we have tried to streamline the descriptions there as well. We have included a separate paragraph describing the results for the wrinkly morphotype GGDEF protein mutant as well.

8. For the *mrkD* mutants, is it known that *mrkA* is expressed at similar levels in these isolates?

Yes, we also measured the *MrkD*^{delG40-43} mutant, and *mrkA* levels can slightly increase in stationary cultures.

9. The Figure 4. c and d legends and the results text need to be swapped around to match the images.

Thank you for noticing. We have now corrected the figure legends.

Reviewer #2 (Remarks to the Author):

Reviewer #3 (Remarks to the Author):

The manuscript uses experimental evolution of three *K. pneumoniae* isolates to identify adaptive changes relevant to biofilm growth in this pathogen. They identify some convergent genetic changes, accompanied by varied phenotypic changes. Some of the genes and pathways that changed during experimental growth overlap with mutations attributed in a separate study to adaptive evolution in vivo during extra-intestinal infections in humans.

General comments:

1. The work presents a large amount of lab experiments, but the statistical analysis does not do them justice. In addition, the data presentation prevents reader from linking different parts of analysis. For example, in Fig 4d, the reader has to mentally align rows for each parental strain to see if same mutation in different strains leads to the same outcome.

Same applies to Fig 4b and 4e and other figures. Statistical tests are used inconsistently. For example, Fig 6e does have any, but Fig 6f does.

We have reviewed the manuscript and ensured that all comparative data are now supported by statistical analysis. We have also updated the Materials and Methods section and figure legends to explain the methodology used. For data presentation, we have chosen to group the mutants with their parental strains when the representation of the phenotypic effects of the respective mutation is best compared to the same genetic background, while we now also compare mutants based on phenotypic characteristics, such as in the new Fig. 4. For comparisons between similar mutations with differences between strains we highlight relevant examples in the text.

2. A lot of conclusions are not backed by any statistical tests which limits their robustness. As a reader, I find it hard to know the samples sizes, distributions, etc. Given the large amount of experimental results, some adjustment for multiplicity of testing is also necessary.

We have now ensured that all comparative data are supported by statistical analysis and updated the Materials and Methods and figure legends to reflect this. Every test that involves multiple comparisons has been adjusted accordingly, and this adjustment has been noted in the descriptions.

Specific comments:

1. Figure 2a: In contrast to the rest of the figures in Figure 2, 2a doesn't split samples by progenitor isolate (IA565, C3091, DA14734). Without this, it's unclear if the change is evenly distributed among clones or due to single clone.

Figure 2 has been modified and no longer includes the previous Figure a.

Lines 98-101: I think this is based on mutation count in Figure B? These mutations are in the core genome (based on Methods), the accessory genome is not taken into account?

There was only one clone from one lineage in the IA565 background that had a mutation in a plasmid-borne gene encoding a hypothetical protein without any conserved domains (DPF99_02100:p.Asp7Gly). We have added a sentence in the Results section to clarify this.

Similarly, "clones had a more diverse mutation spectrum" also requires statistical support, as it's not evident from Fig. 2d.

This section has been updated with statistical analyses.

2. Lines 106: How is the phenotype "hypermucoid" defined?

We have now added a description of the morphotype in the text (Lines 110-114), and further discuss this in the section on mucoviscosity (starting at line 213).

3. Line 107: In the supplementary data (Table S1), the wzc mutations are odd. For

DA14734_full, silicon has 2 lineages with wzc mutations, silicon + fibrinogen has 5, but only two different alleles. HT (low inoculum) has 7 lineages with wzc mutations, but only 5 of them are the same finally HT (high inoculum) has 4 lineages with mutations and three alleles. Using HT (low inoculum) as example, are the authors assuming the mutation arose twice at cycle 4 or seven times at cycle 5? Also, cycle 2 is listed in silicone+fibrinogen, but all other media only show cycle 5 or 6. Does that mean there were no mutations for cycles 1-4?

We have realised that the listing of lineages, clones, and populations was a bit confusing. We have updated Supplementary Table 1 (Now Supplementary Data 2) and the descriptions of how clones and populations were selected for sequencing in the Methods section to clarify this. In short, all lineages were followed through all cycles for morphotype changes; however, for DA14734, not all lineages survived the complete set of cycles. In addition, we sequenced additional clones at cycles where we could identify different morphotypes to understand the evolutionary dynamics better. This results in different numbers of sequenced clones and at different cycles for separate lineages. Therefore, not all lineages have sequences for earlier cycles.

4. Lines 116-117: I'm not sure how these sweeps can be inferred from the cited figures. For Figure 1D, replacing "Morphotype frequency %" with count of samples with morphotypes would be more helpful.

What the figure (now Figure 5) illustrates is not the number of "samples" with morphotypes, but the percent of colonies showing that morphotype in the population in each cycle. Each graph shows a single lineage. For example, 50 hypermuroid colonies out of 100 total colonies would be plotted as 50% hypermuroid frequency. We have added additional descriptions in the figure legend to clarify this. The count of lineages with specific morphotypes for each strain background is now shown in Fig. 1e.

5. Table S1: this table is extremely difficult to understand and appears to have a lot of internal notes. If possible, this table should be cleaned-up as it's very important for understanding the manuscript.

We have cleaned up the table by removing notes and further streamlined the subdivisions of surfaces/lineages/cycles, etc., to make it easier to navigate.

6. Line 136: What is the data for this test?

This is the data on overlapping mutations in outbreak isolates from the previous Figure 3 (now Figure 3 b). We have also included the n numbers in the text.

7. Line 175: Was rcsDG217R was the only pangenome difference between that isolate and parental strain?

Yes, this was the only difference.

8. Fig 4b: missing p-values., annotation for (c) and (d) appears the wrong way round.

Thank you for bringing this to our attention; it has now been corrected.

9. Fig 6e: is this reduction relative to same isolate without cellulase or relative to parental isolate? The implications of this figure on Line 261-264, while accurate, give an impression that these mutations have functional implications, but data doesn't demonstrate this given strain effect and lack of shared mutations in presented strains.

The reduction is relative to the same isolate grown on pegs without cellulase. This has been clarified in the figure legend. Cellulase treatment was done to show the importance of increased cellulose production in mutants with mutations in this system. Mutants that do not rely on cellulose for their increased biofilm formation were unaffected by cellulase.

10. Line 265: The conclusion about mode of mediation can be made if supported either by targeted experiments or statistical model that relates the data in (b), (c), (e) and (f) of Fig. 6.

The role of cellulose is likely to vary between strains and between mutants for several reasons. We have expanded on this in the section and added a comparison between biofilms with and without cellulase treatment for all strains, now in Supplementary Fig. 5.

11. Line 279-280: The data does suggest that cellulase effects biofilm formation, but the inconsistency of results in Fig 6e demonstrates that cellulase does not always inhibits biofilm formation.

Yes, what we find is that some of the mutants have upregulated cellulose production, and this is likely the cause of the increased biofilm capacity, as they are sensitive to cellulase. Other mutants have altered the production of other ECM components or have structural changes in fimbria and are less sensitive to cellulase. This has been clarified in the text.

12. Lines 440-441: How many mutations were accumulated in each cycle in each lineage for different parental strains? This can be demonstrated via a box and whiskers plot with a series for each parental strain and whiskers showing range of mutations across lineages at each cycle. Evidence for Line 497 could also be added to this plot.

We are unsure what the reviewer is aiming for with this calculation, which is not already part of the calculation of the accumulated number of mutations on different surfaces from the endpoint population sequencing (Figure 2). This would require additional population sequencing of all lineages at all timepoints. To track the evolutionary trajectories during evolution, we instead monitored changes in morphotypes across all lineages at all passages and sequenced both populations and isolated clones from representative lineages at multiple time points. We have updated the Materials & Methods section to clarify this. The pilot experiment described in line 497 was performed to verify that a short growth for some lineages to produce enough cells for WGS did not change the population structure.

13. Lines 507-510: I cannot decipher what this means "For clones, the cut-off for calling SNPs was set to >75%, and for InDels, a frequency of >0,5 among reads was used, and each change was manually inspected." or why different cut-offs were used; or why the thresholds are presented by a mix of % and proportion in the same sentence (75% vs 0.5). Please

clarify.

These settings relate to the variant detection tools in the CLC Genomic Workbench. SNPs are relatively easy to detect from short read sequences as they are distinct differences at a single (mostly) nucleotide position compared to the reference sequence. Here, the primary concern is filtering out sequencing errors. However, using a 100% cutoff criterion (all reads have to have the SNP) would risk missing actual SNP calls, as sequencing errors overlapping with the SNP position would lower the % of reads with the SNP. A 75% cutoff allows for the detection of SNPs even in such regions. InDels are inherently trickier to detect since they represent larger differences compared to the reference sequence, and reads from these regions cannot always be placed correctly by the assembly algorithm at the same rate. This always results in reads being excluded by the program, and the cutoff to detect variations must be lowered to avoid missing true differences. Here, short-read sequencing methods that include PCR and ligation steps (such as Illumina) often have a high frequency of reads that contain artificial insertions and deletions (InDels) and therefore require filtering. The use of % and proportion in the same sentence is due to the use of these by the program. We have updated the description to clarify this and harmonized the use of "%" for both calls.

Reviewer #4 (Remarks to the Author):

The group of Sandegren conducted a study investigating the evolution of biofilms in three strains of *Klebsiella pneumoniae*, "simulating" catheter-associated infections through surface-attached biofilms. They observed rapid evolutionary changes affecting capsule formation, fimbrial adhesion, and c-di-GMP-dependent pathways, with results sometimes varying among strains. The study also found that acute virulence was associated with specific genetic changes rather than overall biofilm capacity. Additionally, the authors noted overlaps between the changes and previously identified adaptations in urinary tract and wound isolates. Although the study's goals are clearly outlined and its presentation is satisfactory, there are certain issues that need to be addressed before it can be published in Nature Communications.

Major points

- My primary concern is the limited variety of strains and strain diversity in the study. It would significantly enhance the study's relevance to include a broader range of strains, such as hypervirulent or convergent representatives, to yield more meaningful biological outcomes, especially in the context of capsular adaptations. While the authors did find mutational overlaps with clinical strains and changes documented in the literature, expanding the strain selection would strengthen the study's overall impact and applicability. With that being said, if I understand correctly, the overlap with previous isolates pertain to strains from a clonal outbreak, which typically exhibit comparable genetic compositions. Therefore, it's possible that the number of identical mutations is biased.
- The novelty of both the findings and the applied methodologies seems somewhat limited. Many of the observed changes, as acknowledged by the authors, have been previously documented and are expected. Additionally, the FlexiPeg device has been previously published. As mentioned before, to increase interest, it would be beneficial to include other genetic backgrounds of *K. pneumoniae*, enabling a more comprehensive exploration of commonly applicable patterns. The study could have also been elevated to the next level by

investigating the combination of mutations leading to various phenotypes and exploring the underlying mechanisms of compensatory processes.

While we agree that expanding the strain panel could offer broader biological insights, our study was intentionally focused on the evolutionary dynamics of classical *K. pneumoniae* (cKp), which represent the predominant cause of UTIs and other opportunistic infections. Our aim was to investigate how biofilm capacity evolves specifically within this clinically relevant group. Including hypervirulent or convergent strains would indeed increase diversity, but it would also introduce confounding variables due to their distinct evolutionary trajectories and pathoadaptive profiles. These strains can be considered as already selected variants, shaped by different pressures than those acting on cKp. Rather than incorporating them directly into the experimental evolution, we chose to relate our findings to previously described changes in such isolates, allowing for meaningful comparisons without compromising the clarity of our experimental design.

Regarding the concern about limited novelty, we respectfully note that while some mutational targets have been previously reported, our study provides a unique opportunity to compare experimental evolution with in-host evolution in a clonal outbreak strain. This direct comparison within the same genetic background enables us to interpret specific mutations—such as those in *wzc* and the wrinkly phenotype—with greater resolution and confidence, avoiding ambiguity due to potential epistatic effects. To complement this, we included two well-characterized reference strains to assess the generalizability of our findings across different genetic backgrounds.

As for the FlexiPeg system, while it has been previously published as a methodological tool, its application here for long-term experimental evolution in biofilms represents a novel use case that enables controlled and reproducible selection of biofilm-adapted variants.

We also agree that further exploration of mutational combinations and compensatory mechanisms would be valuable. These investigations are ongoing for selected targets (e.g., *spyo*), and we intend to present them in future studies to maintain a focused narrative in the current manuscript.

Medium points

- The degree to which the device mimics the actual clinical situation should be elucidated further. How closely does the device replicate the clinical scenario?

This is a very important question. Our study aims to replicate the surface, not the whole clinical scenario. Silicone is a standard material for indwelling medical devices, and fibrinogen has been shown in multiple studies to coat the inserted catheters. Although it is impossible to quantify how well a model replicates reality fully, the overlap in mutation spectra between the *in vitro* evolution in the system and the mutations isolated from catheterized patients during the hospital outbreak with the exact bacterial clone strongly suggests that the setup replicates the in-host evolution on the surface. The only group of mutations found in the outbreak that we did not find in the experimental evolution were those involving iron metabolism, which indicates that they were selected by a different

pressure in vivo. We have expanded on this throughout the manuscript when discussing the overlaps.

- L. 103-104: potential trade-offs should be addressed further, e.g. in additional fitness/virulence assays.

Many of the experiments described later in the manuscript assess differences and trade-offs between the mutants (serum sensitivity, Galleria infection potential etc). We have now also added head-to-head competitions between selected mutants in biofilm and planktonic conditions (Fig. 5) to further assess competitive differences between mutants under different growth conditions.

Minor points

- L. 34-35: this is not true for hypervirulent *K. pneumoniae*

Thank you for pointing this out, we are aware of this and have modified the text to clarify further that the study is solely on the evolution of cKp.

- L. 328: hypervirulent or classic *K. pneumoniae*?

Ernst et al., reported this for multiresistant classical ST258 *K. pneumoniae*. This has been updated in the text.

- L. 433: which medium has been used?

All biofilm growth was done in Brain-Heart Infusion medium (BHI) as stated in the paragraph "Bacterial strains and growth conditions".

- Why sometimes 10 and sometimes 6 individual lineages in the experimental evolution?

The evolutions on different surfaces were conducted at different time points, and the decision to use a different number of lineages was made solely to streamline the second round of evolution. More lineages provide a broader base for determining the diversity of evolution, but also add substantial amounts of samples to handle. We determined that 6 lineages would be sufficient to properly analyze the diversity of evolution on silicone and silicone+fibrinogen, while we still included the results from 10 lineages that were obtained with uncoated pegs.

- I believe that the sedimentation assay is solely referenced in the Materials and Methods section.

Thank you for noticing this. The reference has now been added to the main text too.

- Clarify the rationale behind the relatively short duration of the experimental set-up (48 hours max.).

Based on previous experiments in this model system, the number of CFUs does not increase substantially after 48 hours. This means there is limited further growth and mutation supply, mainly an increase in biomass. We should therefore be able to capture the major selective factors, such as initial attachment and the primary expansion of the main biofilm population within the chosen timeframe. While survival in biofilm could be an interesting selective feature that changes the population structure over time, studying that particular feature of biofilm evolution would substantially increase the experimental time. This could be a feature for future additional studies.

REVIEWER COMMENTS

Reviewer #1 (Remarks to the Author):

I have limited my comments to Reviewer 1 though agree with many of the points raised by the other reviewers, particularly the use of statistics and biological rather than technical replicates to demonstrate the phenotypes identified. I have asked my assistant reviewer to also review the changes.

Authors: We are not sure what the reference to biological vs technical replicates means since no particular experiments were mentioned. We favour biological replicates because they capture the primary type of variation more effectively than technical replicates. We used technical replicates for qPCR to verify the accuracy of each biological replicate, but the data analysis was performed only on biological replicates. The number of biological replicates is indicated in each figure legend.

Point 1 – mutation rates in biofilm and planktonic bacteria

Reviewer: The noteworthy results in this manuscript are predominantly related to the biofilm ecological niche and the propensity for mutation in those bacteria present in biofilms. The increased probability of mutation is not explained, though cell density is one candidate. We are 'left in the dark' on the relative mutation rates of planktonic bacteria – are these equally genetically 'plastic' as the counterpart bacteria present in biofilms. This important would reveal whether the higher mutation rates seen in key biofilm components/proteins e.g. the emergence of hypermucooid and other mutants (mrkJ, mrkD etc) are caused by selective pressure under biofilm growth per se and thus likely to come from factors like 'overcrowding' and competition for nutrients.

Authors: We appreciate that the ecological dynamics within biofilms, including factors such as cell density and nutrient competition, are likely to play a significant role in shaping the emergence of adaptive mutants. The question of whether mutation rates differ between planktonic and biofilm-associated bacteria is indeed important, though it falls outside the scope of our current study.

Our experimental evolution approach was specifically designed to enrich for biofilm-forming variants by repeatedly removing planktonic and loosely attached cells. As such, the mutants we observe—such as those affecting mrkJ and mrkD—are likely the result of selection for traits that enhance biofilm formation, rather than an inherently increased mutation rate within the biofilm environment. These mutations may arise in planktonic cells that subsequently attach and persist, or during biofilm development itself.

Reviewer: The manuscript describes mutations present in bacteria in biofilms. The Reviewer has asked the simple question whether the mutation rate is the same or different from planktonic bacteria. There are relatively straightforward measures of mutation rates involving reconstitution of resistance or metabolic deficiencies in selectable single nucleotide point mutants on defined media, and it is not clear why the authors cannot apply these techniques to measuring the rates in biofilms and when the bacteria are in solution. The constant removal of planktonic bacteria is likely to select for rare bacteria that have lost the ability to adhere through e.g. the Type 3 Mrk fimbriae. I feel that this question should be addressed with experimental data.

Authors: We apologize if our initial response was perceived as avoidance due to difficulties performing the experiments. General mutation rate measurements using fluctuation tests are relatively straightforward, although biofilm growth slightly limits this due to the smaller population size, and comparisons with liquid growth must account for differences in total cell number. Our point was that, regardless of the outcome, it would not affect the main conclusion that strong biofilm phenotypes are rapidly selected for, since the mutants' origin (liquid or biofilm) does not affect that. We have now measured point mutation rates (as the rate of rifampicin resistance, that is primarily through single point mutations) for the three parental strains in planktonic growth and biofilms. Because the population sizes on the pegs for the parental strains were too low to generate reliable data, we increased the surface area and applied the same proportional increase in liquid culture volume to avoid biasing the measurements. We used 24 biologically independent cultures per strain to improve data precision.

Mutation rates were within the expected range, indicating that none of the strains exhibited a mutator phenotype. There was no difference in mutation rates between the strains, or between planktonic growth and biofilm growth. There are limited statistical tests to determine the difference in mutation rate between samples, as explained in Foster (2006) "The estimates of m or μ obtained from fluctuation tests are neither normally distributed nor unbiased; therefore, no matter how many times a fluctuation experiment is repeated, it is not valid to take the mean and standard deviation of the results (Asteris and Sarkar, 1996; Jones et al., 1994; Stewart, 1994).", so here we used the lack of overlap between the 95% confidence intervals as an indication. We have included the findings in Results Lines 132-135 and Supplementary Fig. 3.

Point 2 – structured environments and mutation rates

Reviewer: The authors reference a Nature Comms paper in the discussion (#74) that also saw hypermucooid mutants appear in an evolution experiment, but which used serial passaging of planktonic cells – and these mutants were also better biofilm formers. Hence, in Ref #74m forming a biofilm wasn't a driver for mutagenesis.

Authors: In the study in Ref. 79 (previously Ref. 74), the authors did not perform serial passaging of planktonic cells; instead, cultures were statically grown in 24-well plates to produce “structured environments,” as stated by the authors themselves. Therefore, the argument for selection in biofilms stands as described above, but it is not an increase in mutation rate per se, but rather a function of selection within the biofilm niche.

Reviewer: Evolutionary experiments are very tough to control and while the authors have provided a good explanation to counter the Reviewers comments, if the mutation rate is indeed elevated in biofilms, this is something that should be tested and reported.

From line 429: “Our results suggest that biofilm growth serves as a selective pressure for the emergence of both the hypermucooid and capsule-loss mutants, which we also identified as within-host-selected infection site adaptations during the Uppsala hospital outbreak. In parallel with our work, another study reported the selection of wzc mutants in structured environments, which was achieved by serial passaging in statically incubated 24-well plates.”

In the Nat comms paper (actually Ref. 77), they used the term ‘structure’ to refer to limited dispersal and localised gradients in statically grown cultures. But the cells passaged from this part of the culture is still technically planktonic (free floating) and certainly not from the biofilm. So there was no intentional surface selection or maintenance of attached biomass. This point could be clarified (from line 429 in the manuscript), because the way they've tied in their results it reads that ‘structured environments’ equates to biofilm for someone who hasn't read that cited paper, and it's not – it's a fancy term for planktonic. So from their response I'm unsure how it all argues for selection in biofilms vs increases in mutation rates, because that hasn't been tested.

Authors: We agree with the reviewer that control in evolution experiments is highly dependent on the experimental setup. Since the referenced study was not ours, we have retained the authors' description of the “structured environments”, but added that “How this form of growth relates to cell-cell interactions or to interactions with surfaces was not determined.” to point out that it is not known whether biofilm growth was a factor for selection of the mutants. Lines 439-441.

Point 3 – reads like a thesis

The authors have addressed this issue and tightened the writing of the paper

Point 4 – wild type versus parental

The term parental is appropriate – from sequencing of isolates alone it is impossible to establish what is ‘wild type’ and what is a mutant, where there is opacity about the individual lineages, and the sequences of the isolate(s) from which they were derived.

Authors: We agree with the reviewer and have therefore used the term “parental” throughout the manuscript.

Point 5 – multiple PDEs and DGCs, the authors acknowledge that there is considerable complexity in the bacterial ‘management’ of [c-di-GMP] levels, especially in the signals that activate and regulate the multiple guanylate cyclases and PDEs. The ligands and environmental conditions necessary for the activation of these enzymes remains cryptic for the most part but clearly paints a picture of complexity when [c-di-GMP] is a key component of Mrk expression.

Authors: The varying presence of PDEs and DGCs among K. pneumoniae is indeed adding complexity to the analysis of any processes that rely on c-di-GMP. We have sought to clarify what is relevant to our findings.

Question 1

This is satisfactorily answered.

Question 2

This is satisfactorily answered. The addition of further experimental detail provided in the resubmitted manuscript is welcomed.

Question 3

This is satisfactorily answered and the text updated.

Question 4

Is there some sort of quantitative data that might accompany “yield distinct phenotypes across strains” so that others who repeat these experiments can identify the phenomena?

Authors: The most pronounced differences between mutants in different genetic backgrounds were in mucoviscosity, and we have expanded on the description of how mucoviscosity was quantified by the sedimentation assay and the distinct differences this showed in the different strain backgrounds, Lines 221-228. Sedimentation resistance is the preferred quantitative method to measure mucoviscosity in a standardised way (Khadka, S., Ring, B. E., Pariseau, D. A., & Mike, L. A. (2023). Characterization of Klebsiella pneumoniae extracellular polysaccharides. Current Protocols, 3, e937. doi: 10.1002/cpz1.937). In addition, we have now measured capsule polysaccharides for mutants of the IA565 genetic background using the uronic acid assay (Lines 248-251).

Question 5

This is a fascinating observation but it is not clear whether there was a parental strain from early in the outbreak that had different sequences for the ? 4 changed GGDEFs identified in the outbreak strain. Is it evidence of previous mutation and selection or did it occur during the outbreak?

Authors: The isolate with the most mutations in GGDEF/EAL-domain proteins was DA69557 in the previous study. This isolate was an outlier in the number of mutations relative to the index isolate, with 75 point mutations, compared with a median of 10 per isolate in the total dataset. This makes it harder to determine the evolutionary trajectories, and the isolate had no other closely related “intermediate” isolates within the dataset that could help with such identification.

Question 6 – Mrk expression in static cultures

The authors make a clear and honest point about the selection that can go on for hyper fimbriated isolates that aggregate and therefore are difficult to analyse. If this is clearly stated in the revised manuscript then there this is probably all that can be done.

Question 7 – colony phenotypes

This is satisfactorily answered and the text updated.

Question 8

This is satisfactorily answered and the text updated.

Question 9

This is satisfactorily answered and the text updated.

Reviewer #3

Use of statistics – this should be consistent and the same tests used, where possible.

Authors: We have consulted with an expert in biostatistics, who verified the accuracy of the methods used based on the experiments performed and the collected data. We have therefore not changed the manuscript at this point. If any particular tests seem strange to the reviewer, we are happy to look at them specifically.

Measuring hypermucoid – there are simple tests like uronic acid levels in solution, which can quantitatively determine whether the bacteria are hypermucoid.

Authors: The uronic acid method measures total capsule polysaccharides which is, according to literature (Walker et al., 2019; Mike et al., 2021), not always directly related to mucoviscosity. Sedimentation resistance assay is a standardised method for quantitatively measuring mucoviscosity (Khadka, S., Ring, B. E., Pariseau, D. A., & Mike, L. A. (2023) Characterization of Klebsiella pneumoniae extracellular polysaccharides. Current Protocols, 3, e937. doi: 10.1002/cpz1.937), and this is the method we have used to characterize capsule mutants. However, the uronic acid assay can distinguish between cell-free and cell-bound polysaccharides, which is important for describing the effects of wzc mutations. We have therefore now updated the manuscript with that measurement for the mutants of strain IA565. The other wzc mutants (C3091 and DA14734 strains) are so mucoid with the viscosity of the polysaccharide extracts being so high that they cannot be precipitated in a reproducible manner. The new results show nicely that the cell-free polysaccharide content of the mutants is elevated.

Reviewer #2 (Remarks to the Author):
